# Constraining Gaussian Processes to Systems of Linear Ordinary Differential Equations

**Andreas Besginow**
Department of Electrical Engineering and Computer Science
OWL University of Applied Sciences and Arts
Lemgo, Germany
Institute industrial IT Lemgo, Germany
andreas.besginow@th-owl.de

**Markus Lange-Hegermann**
Department of Electrical Engineering and Computer Science
OWL University of Applied Sciences and Arts
Lemgo, Germany
Institute industrial IT Lemgo, Germany
markus.lange-hegermann@th-owl.de

## Abstract

Data in many applications follows systems of Ordinary Differential Equations (ODEs). This paper presents a novel algorithmic and symbolic construction for covariance functions of Gaussian Processes (GPs) with realizations strictly following a system of linear homogeneous ODEs with constant coefficients, which we call LODE-GPs. Introducing this strong inductive bias into a GP improves modelling of such data. Using smith normal form algorithms, a symbolic technique, we overcome two current restrictions in the state of the art: (1) the need for certain uniqueness conditions in the set of solutions, typically assumed in classical ODE solvers and their probabilistic counterparts, and (2) the restriction to controllable systems, typically assumed when encoding differential equations in covariance functions. We show the effectiveness of LODE-GPs in a number of experiments, for example learning physically interpretable parameters by maximizing the likelihood.

## 1 Introduction

Many real world tasks have underlying dynamic behavior, for example chemical reactions [23], systems in bioprocess engineering [27], or population dynamics [70]. One currently widely debated application are compartmental models in epidemiology [20, 35, 52]. Many such systems are linear or can be decently linearized, such as in control theory [73], biology [16], process engineering [1, §9], or engines [7]. Including prior knowledge in the form of differential equations benefits the model fit and enhances interpretability for a model. Hence, modelling differential equations in Machine Learning (ML) has therefore been the focus of much research in both Gaussian Processes (GPs) (e.g. [38, 30, 51, 25, 63]) and Deep Learning (DL) (e.g. [48, 37, 64, 49, 17]).

The class of probabilistic Ordinary Differential Equation (ODE) solvers allow to estimate the solution of ODE initial value problems through approximations, often based on e.g. Runge-Kutta methods or Kalman filters [52, 54, 53, 10], and thereby do not strictly guarantee to yield solutions of the ODE. This class of algorithms is commonly used to solve non-linear ODEs, but typically require that systems are well-posed, in particular the solution needs to be unique once a finite number of

initial conditions is known. While this second limitation is typically irrelevant in physical or biological systems, it is strongly relevant in systems in engineering, such as in control systems with their freely choosable inputs.

With early works like [26], the introduction of derivatives as linear operators in GPs became prominent for partial differential equations. This inspired several works that encode differential equations in the covariance structure of GPs [30, 51, 25, 63, 47] to introduce a strong inductive bias into GPs. This motivated [38] to make these approaches algorithmic, build a mathematical foundation, and show the hidden assumption in previous works of being limited to controllable systems.

This paper overcomes the necessity of approximations, the restriction to systems where initial conditions lead to unique solutions, and the restriction to controllable systems. For that purpose, we algorithmically construct LODE-GPs (Theorem 1), a novel class of GPs whose realizations strictly follow a given system of homogenuous linear ODEs with constant coefficients.

Our construction of LODE-GPs starts by describing our system of ODEs via a so-called operator matrix $A$. Calculating its Smith Normal Form (SNF) gives us a decoupled representation of the system via a diagonal operator matrix $D$, at the cost of going through two invertible base change matrices $U, V$ with the relationship $U \cdot A \cdot V = D$. We can easily construct a multi-output GP $g = \mathcal{GP}(0, k)$ (cf. Lemma 1) with as many outputs as the system channels and whose covariance function encodes the decoupled system. Then, applying the base change matrix $V$ to $g$ yields the pushforward GP $V_* g = \mathcal{GP}(0, VkV')$ whose covariance function encodes the original system, i.e. the LODE-GP. We refer to Theorem 1 and its algorithmic proof for details.

Our paper makes the following contributions:

1. It develops and proves an algorithmic and symbolic construction of LODE-GPs, GPs with the strong inductive bias that their realizations strictly satisfy *any* given system of homogeneous linear ODEs with constant coefficients (see Theorem 1).

2. It demonstrates that the constructed GPs are numerically stable and robust (see Section 5).

3. It automatically includes ODE system parameters as GP parameters, learns them during the training process of the GP, and allows an interpretation of the data (see Subsection 5.2).

4. It provides a free and public implementation based on the GP library GPyTorch[1] [22].

We successfully test our approach on controllable and non-controllable systems of differential equations, with and without system parameters. The LODE-GP strictly satisfies the ODEs and outperforms GPs by several magnitudes in its precision, additionally it correctly reconstructs the system parameters used to generate the data with a small relative error, even despite noise. We also discuss how this strict behavior influences results for metrics like the RMSE with e.g. Figure 4.

## 2 Related Work

Using physical information in ML systems became a central research topic over recent years in both DL and GPs. With the GP research in two important categories: (1) Works that introduce derivative information as inductive bias into the GP [30, 38, 39, 26, 51, 25, 21] and (2) works that train GPs without inductive bias on physical data, with specially selected standard kernels, as an approximate model for additional processing steps [69, 63, 57, 33, 9, 12, 13]. Of all the works on GPs, [38, 39] are the closest to us, as they also provide general algorithms to introduce inductive bias into GPs, based on Gröbner bases. These approaches are only applicable to controllable systems. By basing our algorithm on the SNF instead of Gröbner bases, we overcome this limitation to controllable systems, but are restricted to *ODEs*.

Probabilistic ODE solvers like [52, 34, 62, 10, 42, 53, 54, 8, 12] are able to include inductive bias of ODEs but sometimes require a number of comparably complex calculations and some approximations. We avoid any approximation through calculation of the SNF and the application of the pushforward with a linear differential operator. Hence, our work is limited to *linear* ODEs, whereas probabilistic ODE solvers are able to work with non-linear ODEs.

For GP priors for decoupled ODE systems with constant coefficients and right hand side functions see [2], which considers the right hand side functions as latent, hence these models are called Latent

---

[1] https://github.com/ABesginow/LODE-GPs

Force Model (LFM). A GP prior on these latent forces and is pushed forward through differential operators and Green's operator. We empirically compare LODE-GPs to LFMs in the appendix.

Of the other works that introduce inductive bias, [26] and [51] are the earliest works and already discuss a general formulation of differential equations by viewing derivatives as linear operators that can be applied to GPs. Subsequent works target specific ODEs and create covariance functions with inductive bias [25, 63, 15]. They can be considered special cases of our approach with a specific manually constructed pushforward. Finally, others constructed covariance functions for *partial* differential equations from standard covariance functions with e.g. curl-free behaviour [69] to model physical behaviour [69, 63, 57, 33] or use the GP as a surrogate model to learn physical problems like the inverse pole problem [13] or a variety of applications [9].

Most DL models explore physically informed models through additional loss terms that punish solutions that deviate from the system [64, 37, 48], or in the case of [17] show the equivalence of the loss to their assumed physical behavior. This is often combined with different modifications to the networks like additional features [17] or specific network structures [37].

# 3 Background

## 3.1 Gaussian Processes

A GP $g = \mathcal{GP}(\mu, k)$ defines a probability distribution over the space of functions $\mathbb{R}^d \to \mathbb{R}^\ell$, such that the outputs $g(x_i)$ at any set of $x_i \in \mathbb{R}^d$ are jointly Gaussian [72]. Such a (multi-output) GP is defined by its mean function (often set to zero)

$$\mu : \mathbb{R}^d \to \mathbb{R}^\ell : x \mapsto \mathbb{E}(g(x))$$

and its (multi-output) positive semi-definite covariance function (also called kernel)

$$k : \mathbb{R}^d \times \mathbb{R}^d \to \mathbb{R}^\ell_{\succeq 0} : (x, x') \mapsto \mathbb{E}((g(x) - \mu(x))(g(x') - \mu(x'))^T).$$

A popular kernel is the Squared Exponential (SE) kernel $k_{SE}(x, x') = \sigma^2 \exp\left(-\frac{(x-x')^2}{2\ell^2}\right)$, with its signal variance $\sigma$ and lengthscale $\ell$. It models smooth data very well and is thus usable for many datasets [72, p. 83], as its realizations are dense in the space of smooth, i.e. infinitely differentiable, functions $C^\infty(\mathbb{R}, \mathbb{R})$ [40, Prop. 1].

Manipulating existing GPs allows to introduce indutive biases in various ways [18, 19, 56, 11, 51]. One important example is the application of matrices of linear operators to a GP. Formally, this is the pushforward operation of an operator matrix $B$ on the GP $g$ as $B_* g = \mathcal{GP}(B\mu(x), Bk(x, x')(B')^T)$ [4], where $B'$ denotes the operation of $B$ on the second argument of $k(x, x')$ [39, Lemma 2.2]. These operators can, for example, be differential operators. The matrix $B$ induces the strong bias such that all realizations lie in the image of $B$. This pushforward is typical for applications in differential equations [30, 38, 39, 26, 51] or geometry [28].

**Example 1.** GPs can be constrained to realizations satisfying a system of linear equations given by a matrix $A$, e.g. $A = [2 \quad -3]$

$$\mathrm{sol}_\mathcal{F}(A) := \left\{ [f_1(x) \quad f_2(x)]^T \in \mathcal{F}^{2 \times 1} \,\middle|\, A \cdot [f_1(x) \quad f_2(x)]^T = 0 \right\}.$$

with $\mathcal{F} = C^\infty(\mathbb{R}, \mathbb{R})$ the space of smooth input functions. The matrix $B = \begin{bmatrix} 3 \\ 2 \end{bmatrix}$ is maximal in that it solves $A \cdot B = 0$ and hence $\mathrm{sol}_\mathcal{F}(A) = B \cdot \mathcal{F} = \{B \cdot f(x) \,|\, f(x) \in \mathcal{F}\}$. Taking a GP prior $g = \mathcal{GP}(0, k)$ for $f(x) \in \mathcal{F}$ and applying the pushforward then yields a new, constrained GP prior

$$B_* g = \mathcal{GP}(0, Bk(B')^T) = \mathcal{GP}\left( \begin{bmatrix} 0 \\ 0 \end{bmatrix}, \begin{bmatrix} 9 \cdot k_{SE} & 6 \cdot k_{SE} \\ 6 \cdot k_{SE} & 4 \cdot k_{SE} \end{bmatrix} \right)$$

for $B \cdot f(x) \in \mathrm{sol}_\mathcal{F}(A)$, whose realizations are guaranteed to lie in $\mathrm{sol}_\mathcal{F}(A)$.

This example is similar to our use case of systems of differential equations for LODE-GPs, where we just replace the matrix $B$ of numbers with a suitable matrix of differential operators.

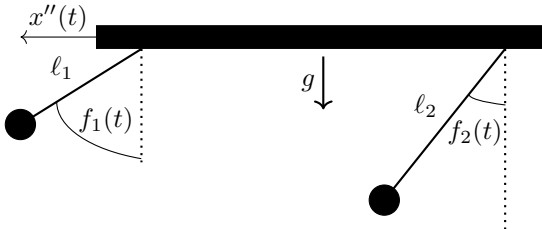

Figure 1: A visualization of the bipendulum with its components (rope lengths $\ell_1, \ell_2$) and states (angles $f_1(t), f_2(t)$, rod acceleration $x''(t)$).

## 3.2 Smith normal form

The SNF [55, 44] is a normal form for a matrix $A \in \mathbb{R}[x]^{m \times n}$ over the polynomial ring $\mathbb{R}[x]$, such that $U \cdot A \cdot V = D$. Here, $D \in \mathbb{R}[x]^{m \times n}$ is a (not necessarily square) diagonal matrix of same size as $A$ and the base change matrices $U \in \mathbb{R}[x]^{m \times m}$ and $V \in \mathbb{R}[x]^{n \times n}$ are invertible square matrices, i.e. $\det(U), \det(V) \in \mathbb{R} \setminus \{0\}$ [24, 14]. Algorithms to construct the SNF are implemented in many computer algebra systems such as Matlab [43], Maple [41], or SageMath [60] (based on PARI [59]) which is a free and open source python library for computer algebra. Intuitively, the SNF can be compared to the eigendecomposition of a $\mathbb{R}^{n \times n}$ matrix, which—if it exists—produces a square matrix of eigenvectors $W$ and the diagonal matrix of eigenvalues $\Lambda$. This analogy is lacking since e.g. the SNF produces two independent base change operator matrices $V$ resp. $U$ instead of a single eigenvector matrix $W$ resp. its inverse $W^{-1}$. Still, the matrix $D$ decouples and thereby simplifies a system given by the input matrix $A$, as does the diagonal eigenvalue matrix $\Lambda$ [50, 14].

Since the SNF exists for matrices over polynomial rings over any field, we can also compute it over a polynomial ring over the function field $\mathbb{R}(a_1, \ldots, a_k)$. Hence, we can model differential equations containing parameters $a_1, \ldots, a_k$. The heating example in Subsection 5.2 demonstrates how such parameters are being used in the SNF and subsequently learned from data.

**Example 2.** A bipendulum acts as our running example. It consists of a rod with a pendulum attached to each end of the rod, see Figure 2. The linearized equations of an idealized bipendulum:

$$\begin{bmatrix} x''(t) + \ell_1 f_1''(t) + g f_1(t) \\ x''(t) + \ell_2 f_2''(t) + g f_2(t) \end{bmatrix} = \mathbf{0} = \underbrace{\begin{bmatrix} \partial_t^2 + \frac{g}{\ell_1} & 0 & -\frac{1}{\ell_1} \\ 0 & \partial_t^2 + \frac{g}{\ell_2} & -\frac{1}{\ell_2} \end{bmatrix}}_{A} \cdot \begin{bmatrix} f_1(t) \\ f_2(t) \\ u(t) \end{bmatrix} \tag{1}$$

with the operator matrix $A$, $\ell_i > 0$ the length of the $i$-th pendulum, $g = 9.81$ the gravitational constant, $u(t) = -x''(t)$ the rod's acceleration, and $f_i(t), i = 1, 2$ the angle between the $i$-th pendulum and the vertical axis. The SNF of $A$ reveals its properties, specifically when $\ell_1 = \ell_2$ and $\ell_1 \neq \ell_2$. An algorithm calculates the SNF $D$ of $A$ and the base change matrices $U, V$ as follows:

case $\ell_1 = 1 \neq \ell_2 = 2$:

$$\underbrace{\begin{bmatrix} 1 & 0 \\ -\frac{1}{2} & 1 \end{bmatrix}}_{U} \underbrace{\begin{bmatrix} \partial_t^2 + g & 0 & -1 \\ 0 & \partial_t^2 + \frac{g}{2} & -\frac{1}{2} \end{bmatrix}}_{A} \underbrace{\begin{bmatrix} 0 & -\frac{4}{g} & \frac{2\partial_t^2 + g}{2} \\ 0 & -\frac{2}{g} & \frac{\partial_t^2 + g}{2} \\ -1 & -\frac{4\partial_t^2 + 4g}{g} & (\partial_t^2 + \frac{g}{2})(\partial_t^2 + g) \end{bmatrix}}_{V} = \underbrace{\begin{bmatrix} 1 & 0 & 0 \\ 0 & 1 & 0 \end{bmatrix}}_{D}$$

case $l_1 = l_2 = 1$:

$$\underbrace{\begin{bmatrix} 1 & 0 \\ -1 & 1 \end{bmatrix}}_{U} \underbrace{\begin{bmatrix} \partial_t^2 + g & 0 & -1 \\ 0 & \partial_t^2 + g & -1 \end{bmatrix}}_{A} \underbrace{\begin{bmatrix} 0 & 0 & 1 \\ 0 & 1 & 1 \\ -1 & 0 & \partial_t^2 + g \end{bmatrix}}_{V} = \underbrace{\begin{bmatrix} 1 & 0 & 0 \\ 0 & \partial_t^2 + g & 0 \end{bmatrix}}_{D}$$

In case of unequal rope length, the controllability is intuitive when imagining that, by accelerating the rod at the right time, the two ropes can reach every combination of valid positions. Formally,

this follows that $D \cdot [h_1(t) \quad h_2(t) \quad h_3(t)]^T = 0$ if and only if $h_1(t) = h_2(t) = 0$ and $h_3(t)$ is an arbitrarily choosable output. In case of equal rope length, the system is not controllable, indicated by the diagonal entry $\partial_t^2 + g$ of $D$ [45, p. 154].

## 4 Constructing a GP for differential equations

We introduce a polynomial time algorithm to construct LODE-GPs, a class of GPs with realizations dense in the space of solutions of linear ODEs with constant coefficients. This ensures that the LODE-GP is able to produce all possible solutions to the ODEs and nothing but solutions for the ODEs. All this is guaranteed to be strictly accurate, since we make no approximations.

Consider a system of linear homogenous ordinary differential equations with constant coefficients

$$A \cdot \mathbf{f}(t) = 0 \tag{2}$$

with operator matrix $A \in \mathbb{R}[\partial_t]^{m \times n}$ determining the relationship between the smooth functions $f_i(t) \in C^\infty(\mathbb{R}, \mathbb{R})$ of $\mathbf{f}(t) = (f_1(t) \quad \dots \quad f_n(t))^T$. For such systems our main result holds.

**Theorem 1.** *(LODE-GPs) For every system as in Equation (2) there exists a GP g, such that the set of realizations of g is dense in the set of solutions of $A \cdot \mathbf{f}(t) = 0$.*

The following lemma play a crucial role in the proof of this theorem by constructing base covariance functions for a system as in Equation (2) that is *decoupled* (via the SNF) into scalar equations. The proof in Appendix D makes use of the solution space being decomposed, in particular it deals with finite dimensional vector spaces like Bayesian linear regression.

**Lemma 1.** *Covariance functions for solutions of the scalar linear differential equations $d \cdot f = 0$ with constant coefficients, i.e. $d \in \mathbb{R}[\partial_t]$, are given by Table 1 for d is primary, i.e. a power of an irreducible real polynomial. In the case of a non-primary d, each primary factor $d_i$ of $d = \prod_{i=0}^{\ell-1} d_i$ is first translated to its respective covariance function $k_i$ separately before they are added up to give the full covariance function $k(t_1, t_2) = \sum_{i=0}^{\ell-1} k_i(t_1, t_2)$.*

Table 1: Primary operators $d$ and their corresponding covariance function $k(t_1, t_2)$.

| $d$ | $k(t_1, t_2)$ |
|---|---|
| 1 | 0 |
| $(\partial_t - a)^j$ | $\left( \sum_{i=0}^{j-1} t_1^i t_2^i \right) \cdot \exp(a \cdot (t_1 + t_2))$ |
| $((\partial_t - a - ib)(\partial_t - a + ib))^j$ | $\left( \sum_{i=0}^{j-1} t_1^i t_2^i \right) \cdot \exp(a \cdot (t_1 + t_2)) \cdot \cos(b \cdot (t_1 - t_2))$ |
| 0 | $\exp(-\frac{1}{2}(t_1 - t_2)^2)$ |

For the case of single (i.e. $j = 1$) zeroes, the sum in the covariance function simplifies to a factor 1.

**Example 3.** The ODE $d = \partial_t - 1$ has solutions of the form $a \cdot \exp(t)$ for $a \in \mathbb{R}$. This one-dimensional solution space is described by the covariance function $k(t_1, t_2) = \exp(t_1) \cdot \exp(t_2) = \exp(t_1 + t_2)$.

*Proof of Theorem 1.* We begin with a neutral multiplication of $A \cdot \mathbf{f} = 0$ with $V \cdot V^{-1}$ and a left multiplication by $U$ to decouple the system using the SNF $D = U \cdot A \cdot V$ (cf. Equation (2)):

$$
\begin{aligned}
& U \cdot A \cdot V \cdot V^{-1} \cdot \mathbf{f} = 0 \\
\Leftrightarrow \quad & D \cdot V^{-1} \cdot \mathbf{f} = 0 \\
\Leftrightarrow \quad & D \cdot \mathbf{p} = 0 \\
\Leftrightarrow \quad & \bigwedge_{i=1}^{\min(n,m)} D_{i,i} \cdot \mathbf{p}_i = 0 \quad \wedge \quad \bigwedge_{i=\min(n,m)+1}^{n} 0 \cdot \mathbf{p}_i = 0
\end{aligned} \tag{3}
$$

for decoupled latent vector $\mathbf{p} = V^{-1}\mathbf{f}$ of functions.

We assume the GP-prior $h \sim \mathcal{GP}(\mathbf{0}, k)$ for this vector $\mathbf{p}$ via a GP, where $k$ is a multi-output diagonal covariance function such that each diagonal entry of $k$ is given by Lemma 1. By this Lemma, the set of realizations of this prior $h$ is dense in the set of solutions of $D \cdot \mathbf{p} = 0$.

The pushforward of $h$ with $V$ yields a GP $g$ that can learn the original system of ODEs.

$$g \sim V_* h = \mathcal{GP}(\mathbf{0}, V \cdot k \cdot V') \tag{4}$$

with $V' = V^T$ the operation applied on the second entry of the kernel (i.e. $t_2$) using [39, Lemma 2.2]. As $V$ is invertible, the set of realizations of $g$ is dense in the set of solutions of $A \cdot \mathbf{f} = 0$. $\qquad\square$

This proof shows that diagonalizing the system to $D \cdot \mathbf{p} = 0$ decouples the components of $\mathbf{p}$ to expose the intrinsic behavior of the system. We can also easily interpret the controllability of a system in this decoupled form. A zero column (or $0 \cdot \mathbf{p}_i = 0$) stands for a freely choosable function in $\mathbf{p}$, i.e. an output of the system. A one in a column (or $1 \cdot \mathbf{p}_i = 0$) stands for something we have no choice in, e.g. an input or a state that is fixed once the desired output was chosen. Any other entry leads to sinusoidal-exponential behavior, which is uncontrollable in the sense that it cannot be influenced. This behaviour is classic in systems of differential equations, as most of them can be split into a controllable and uncontrollable solution space, which can be clearly seen in the entries of $D$. Summing up, a system is controllable iff all diagonal entries of $D$ are either zero or one.

This paper improves upon [38] for two reasons: first, it decouples the uncontrollable part of a system via the SNF as a direct summand and, second, it constructs a covariance function for this uncontrollable summand. This is generally impossible for partial differential equations due to [3, 46].

The runtime of the construction of LODE-GPs in Theorem 1 is dominated by computing the SNF. This has a deterministic polynomial complexity in the size of the (usually small) operator matrices $A$ [65, 67, 36, 71] with quicker probabilistic Las Vegas algorithms [58], and it can be parallelized [66, 68]. For our examples, the SNF terminates instantaniously. The application of the base change matrix $V$ is cubic in the (usually small) size of matrices, assuming constant time to apply operators to the covariance functions from Lemma 1. Once the covariance function is constructed, the complexity of $\mathcal{O}(n^3)$ for GPs applies, where $n$ is the (potentially big) number of data points.

**Example 4.** We continue Example 2 and consider the SNF of $A$ for $l_1 \neq l_2$. Since the three diagonal entries of the matrix $D$ are $1, 1, 0$ we conclude that latent kernel $k$ has diagonal entries $0, 0, \mathrm{k_{SE}}$ with Lemma 1. Now the pushforward creates the LODE-GP, which requires the following application of two operator matrices; we simplify by removing irrelevant zeroes of the latent covariance.

$$V k V' = \begin{bmatrix} \frac{2\partial_{t_1}^2 + g}{2} \\ \frac{\partial_{t_1}^2 + g}{2} \\ (\partial_{t_1}^2 + \frac{g}{2})(\partial_{t_1}^2 + g) \end{bmatrix} \cdot [\mathrm{k_{SE}}] \cdot \begin{bmatrix} \frac{2\partial_{t_2}^2 + g}{2} & \frac{\partial_{t_2}^2 + g}{2} & (\partial_{t_2}^2 + \frac{g}{2})(\partial_{t_2}^2 + g) \end{bmatrix}$$

Applying these two operators to the kernel $\mathrm{k_{SE}}$ results in the covariance function for the bipendulum.

## 5    Experimental evaluation

We demonstrate the effectiveness of LODE-GPs in three examples, where we constrain a LODE-GP using the system description and train it with datapoints, similar to solving an initial value problem. The Gröbner basis approach of [38] yields precisely the same results for the two controllable examples and is not applicable to the uncontrollable example. The only other method that could deal with our class of differential equations is [2][2], for which a comparison with the three tank system is discussed in the appendix, in the following we compare our model mainly to classic GPs.

Our comparison includes the error in satisfying the ODEs, specified by the median error the GPs posterior mean function has in satisfying the ODEs at evenly spread points, where we calculate derivatives through finite differences. The LODE-GPs hyperparameters (lengthscale and signal variance) are randomly initialized from a uniform distribution on $[-3, 3]$ and were trained using Adam [32]. The 25 training datapoints are created uniformly in the intervals $[1, 6]$ and $[-5, 5]$ for the bipendulum and three tank system, resp. the heating system. Training and evaluation is repeated

---

[2]Most probabilistic ODE solvers are not applicable to systems with free functions in their solution set, with the exception of [52], which can only estimate free functions of parameters.

Table 2: The median results and standard dev. for training and evaluation RMSE, loss value (negative marginal log likelihood per datapoint) and the mean ODE satisfaction error, over 20 experiments for the GP and LODE-GP on noisy training data. The mean ODE error is the average of each ODE error for a function in a system. The RMSEs were calculated with noiseless data in the training and evaluation interval, the ODE error only with the latter, for each respective experiment. Smaller is better.

| | | training RMSE | evaluation RMSE | loss | mean ODE error |
|---|---|---|---|---|---|
| Bipendulum | LODE-GP | $0.060 \pm 0.060$ | $0.106 \pm 0.258$ | $-1.638 \pm 0.567$ | **5.135e-07** $\pm$ 1.607e-06 |
| | GP | **0.008** $\pm 0.030$ | **0.044** $\pm 0.0175$ | **-2.072** $\pm 0.215$ | $0.064 \pm 0.023$ |
| Heating | LODE-GP | **0.006** $\pm 0.001$ | **0.170** $\pm 0.068$ | **-2.089** $\pm 0.100$ | **0.008** $\pm 0.007$ |
| | GP | $0.010 \pm 0.001$ | $0.333 \pm 0.055$ | $-1.628 \pm 0.051$ | $0.218 \pm 0.070$ |
| Three tank | LODE-GP | **0.023** $\pm 0.004$ | $0.078 \pm 0.046$ | $-0.949 \pm 0.063$ | **1.360e-05** $\pm$ 4.919e-06 |
| | GP | $0.028 \pm 0.005$ | **0.058** $\pm 0.031$ | **-0.974** $\pm 0.071$ | $0.040 \pm 0.020$ |

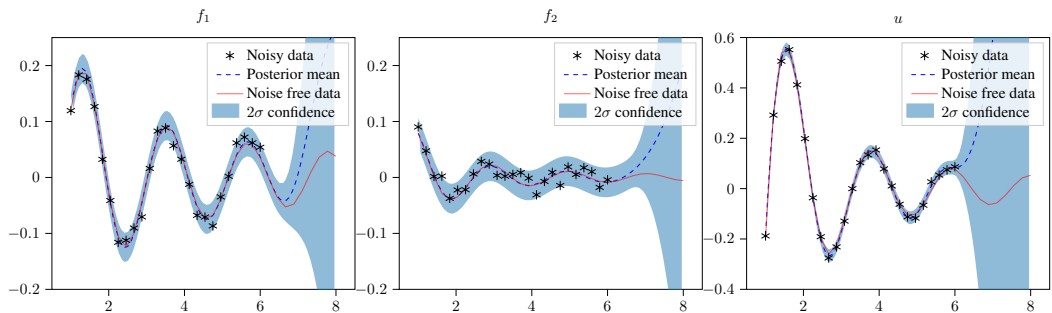

Figure 2: A posterior LODE-GP trained on noisy bipendulum data (black stars), its confidence (blue transparent), posterior mean (blue dashed line) and the original noise-free function (red line). The posterior mean fits the original data very well and shows that the behavior is learned despite noise.

ten times in each experiment using a GPyTorch [22] implementation of our LODE-GP construction with SageMath [60] to symbolically calculate the SNF. The training loss is the negative marginal log likelihood (c.f. [72, Eq. (2.30)]) which, in GPyTorch, is additionally divided by the number of training data points. Details of the training, data generation, verification process, additional experiments, e.g. without noise, and further system details can be found in the appendix.

## 5.1 Bipendulum - Controllable

We continue with the controllable variant of the bipendulum from Example 2 with the rope lengths $\ell_1 = 1$ and $\ell_2 = 2$. We create 25 points of training data from a solution of the ODEs as shown in Figure 2. We add white noise to the data with standard deviation of 2% of the maximal signal.

Figure 3 compares the error in satisfying the ODE of a LODE-GP and a GP. The LODE-GP is producing samples that satisfy the ODE with a median error[3] of 2e−6, for both differential equations. In comparison, the *symbolically computed solution* used to generate the training data, has an error of 1.75e−7, only one order of magnitude better than the LODE-GP. Hence, we conclude that, up to numerical precision, the LODE-GP produces samples that strictly satisfy the ODE, whereas regression models like GPs do not satisfy the differential equations. The LODE-GP produces roughly 30000 times more precise samples (0.06 to 2e−6) and deviates only slightly from this error.

Further we investigate the RMSE of the models by comparing their posterior mean to noiseless data in the training and evaluation intervals. The LODE-GP RMSE mostly performs similar compared to the GP RMSE (see Table 2). For the training interval, the LODE-GP achieves basically the

---

[3]The LODE-GPs trains into a local minimum at its value $10^{-10}$, with high lengthscale and miniscule signal variance. This models constant zero behaviour which satisfies the ODEs, but does not fit the training data.

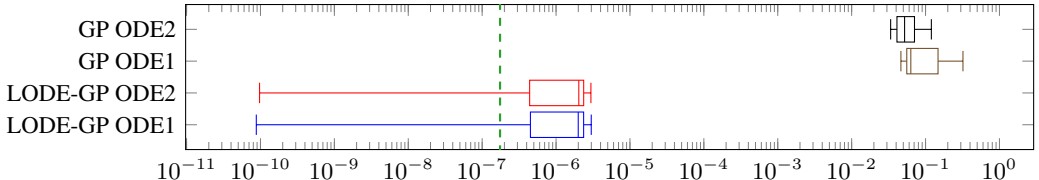

Figure 3: The error in satisfying the bipendulum ODEs in Equation (1) for a GP (top) and LODE-GP (bottom), for noisy training data. Smaller is better. The green dashed line shows the error of a solution in symbolic form. The low error shows that the LODE-GP strictly satisfies the ODEs, up to numerical precision.

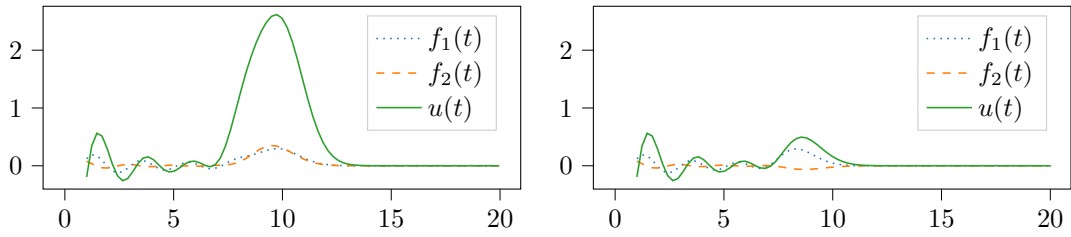

Figure 4: The posterior mean functions of a LODE-GP (left) and a GP (right) trained on the bipendulum data as described in the text.

same performance as the GP. For the evaluation interval the performance worsens due to a SE GP extrapolating to its (constant zero) prior mean. The GP can easily reach this mean zero, by smoothly modifying all channels without regard for the system equations. Since the ODE solution also approaches zero, this leads to a small error in extrapolation. The LODE-GP has to satisfy the system equations to move the mean function back to mean zero. Hence, the system produces a large spike[4] in the input channel $u(t)$, before the system can reach mean zero, as seen in Figure 4.

We also analyse the training runtime and see that the LODE-GP is as fast as the GP during the $\mathcal{O}(n^3)$ model inference and slower by a factor smaller than two in the $\mathcal{O}(n^2)$ covariance matrix calculation.

## 5.2 Heating system - Controllable with parameters

In this experiment we use a parameterized heating system to show that a LODE-GP can learn physically interpretable system parameters during GP hyperparameter training. The controllable heating system is depicted in Figure 5. It uses an input $u(t)$ to control a heating element $f_1(t)$ which exchanges heat with an object $f_2(t)$. Two physical parameters $a$ and $b$ determine how quickly the heat moves between the objects according to the ODEs:

$$f_1'(t) = -a \cdot (f_1(t) - f_2(t)) + u(t)$$
$$f_2'(t) = -b \cdot (f_2(t) - f_1(t))$$

We generate training data using a solution of the differential equation (see Figure 5) with parameter values $a = 3, b = 1$. The LODE-GP can be constructed with parameters in the ODEs.

Learning the physical parameters $a$ and $b$ reconstructs the original values successfully with a maximal relative error of less than 2.8% in ten training runs on data without noise. After adding white noise to the data with standard deviation of 1% of the maximal signal, the parameters were successfully reconstructed with a maximal relative error of 5.3% in ten training runs. Again, the LODE-GP satisfies the original ODEs (with parameters $a = 3$ and $b = 1$) with an median error of $1e{-}2$, trained on the noisy data, we refer the reader to Appendix B.3 for a visualization similar to Figure 7. This is bigger than in the previous example, as the LODE-GP only uses approximate parameter values for

---

[4]The third data channel is particularly large due to its additional factor of $g = 9.81$, which has a different scale by an order of magnitude as the otherwise small data.

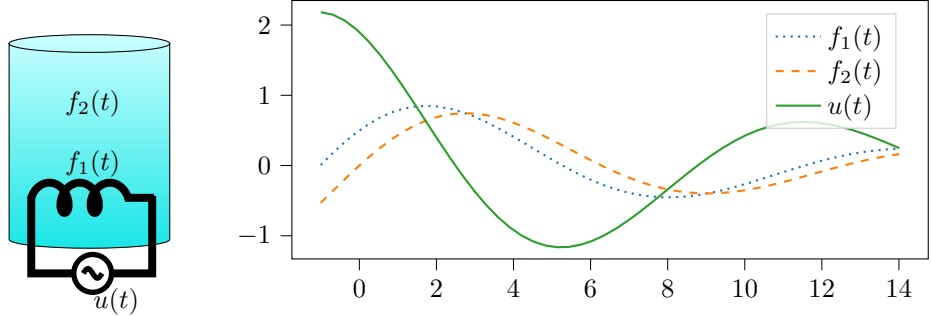

Figure 5: (left) A visualization of the heating system with input $u(t)$ and states $f_1(t), f_2(t)$. (right) A solution of the differential equations, which are used to create the data.

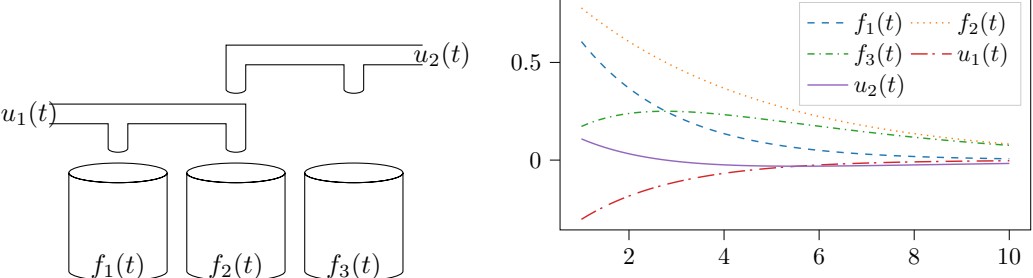

Figure 6: (left) A sketch of the three tank system and (right) a solution of the system.

$a$ and $b$, but still smaller than the error of a GP of $0.20$. Our interpretation of these results is that the LODE-GP can learn the physically interpretable parameters.

### 5.3 Three tank system - Non-controllable

We conclude the experiments with a non-controllable fluid system where the water level in three tanks is changed by two pipes. The system is non-controllable due to pipes' overlap over the center tank, whose changes directly affect the other tanks. This system requires multiple non-zero covariance functions in the latent GP to describe both the non-controllable subsystem and the two degrees of freedom, which corresponds to the columns of the SNFs matrix $D$ via Lemma 1:

$$D = \begin{bmatrix} 1 & 0 & 0 & 0 & 0 \\ 0 & 1 & 0 & 0 & 0 \\ 0 & 0 & -\partial_t & 0 & 0 \end{bmatrix}$$

We use a solution to the system of ODEs (see Figure 6) to generate 25 datapoints, to which we add white noise with standard deviation of 10% of the maximal signal.

The system ODEs are again strictly satisfied by the LODE-GP with a median error of $1e{-}5$ in each ODE. We refer the reader to Appendix B.4 for a visualization similar to Figure 7. Hence, LODE-GP can handle larger, non-controllable systems of ODEs, despite high noise.

## 6 Conclusion

In this paper we have introduced an algorithmical approach to automatically and symbolically create LODE-GP, a class of covariance functions for GPs such that their realizations strictly follow a given system of linear homogeneous ODEs. We have proven that this approach is mathematically sound and verified its effectiveness in experiments. We have demonstrated that the learned posterior mean strictly satisfies the given system of ODEs up to numerical precision and discussed the consequences of this strict behavior e.g. in their extrapolation, leading to drastic behavior when both following the

ODEs and going back to the prior mean. Additionally, we automatically trained the physically interpretable parameters of the system and were able to reconstruct their original values with low relative error. The runtime of the LODE-GP is still asymptotically dominated by the $\mathcal{O}(n^3)$ Cholesky decomposition of the covariance matrix, despite bigger constants in the $\mathcal{O}(n^2)$ part of constructing the covariance matrix and a (usually very small) constant time precomputation of the covariance function. Of course, approximate GPs are applicable to our covariance functions. Beyond applications where the dynamic behavior is known through corresponding ODEs, the combination with kernel search methods [31, 19, 29, 6, 5] has the potential to detect unknown dynamical behavior in data by systematic exploration and use these findings to generate new interpretable knowledge about the data through the corresponding system.

## Acknowledgments and Disclosure of Funding

This research was supported by the research training group "Dataninja" (Trustworthy AI for Seamless Problem Solving: Next Generation Intelligence Joins Robust Data Analysis) funded by the German federal state of North Rhine-Westphalia.

We want to thank the anonymous reviewers for their valuable feedback.

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
