## Appendices

## A    Constructing and training a LODE-GP

The construction of a LODE-GP, as discussed in the proof of Theorem 1, involves the following steps:

1. Calculate the SNF symbolically using SageMath [60]

2. Construct the covariance function for the GP-prior $h \sim \mathcal{GP}(\mathbf{0}, k)$, according to Lemma 1 and the last line of Equation 3

3. Calculate the final LODE-GP through the pushforward with the matrix $V$, including the matrix multiplication $V \cdot k \cdot V'$ and applying the derivatives to $k$

This construction gives us our LODE-GP, which we train using the standard GP training procedure. All SE kernels that are part of the LODE-GPs have randomly initialized signal variance $\sigma$ and lengthscale $\ell$ values chosen uniformly from $[-3, 3]$, which are passed through an $\exp$, during calculation, to ensure numerical stability (e.g. preventing negative lengthscales) and smooth training. We use an Adam optimizer with a learning rate of $0.1$ for $300$ training iterations in all experiments, further we use the GPyTorch multitask likelihood, since the LODE-GP inherits the class of a GPyTorch multitask GP. We reduced the noise constraint for the likelihood from $1e-4$ to $1e-10$. GPyTorch uses the softplus operation to ensure the constraint. We performed calculations with 64 bit precision.

Training and evaluation was done on our server equipped with a Intel(R) Core(TM) i9-10900K CPU @ 3.70GHz and 64GB RAM running at 3200MHz. All calculations were done on the CPU.

Training was done on 25 datapoints and was repeated ten times for noisy and noise-free, for both LODE-GP and GP.

At no point in training or evaluation is any data or GP output scaled, i.e. we use all the data as is.

## B    Additional details on experiments and results

We compare the values for the GP and LODE-GP parameters across the three experiments in Table 3. The values for the parameters are their actual values as used in the calculation, i.e. after applying the softplus and exp functions. The behaviour discussed in Section 5 is clearly visible in the maximum LODE-GP lengthscale for the bipendulum, where it sometimes learns a basically constant function. For the bipendulum and the three tank experiments, the LODE-GP generally learns a higher lengthscale, which translates to a smoother behaviour in the outputs.

Analyzing the eigenvalues for the LODE-GP and GP shows that the LODE-GP produces roughly half as many non-zero eigenvalues (i.e. eigenvalue $\lambda > 1e-6$) compared to the GP, as shown in Table 4. This behaviour is expected since the LODE-GP is underlying additional strong constraints due to the inductive bias, which is reflected in its eigenvalues.

### B.1    Calculating the ODE error

To calculate the error in satisfying the ODEs via finite differences, we generate uniformly distributed datapoints in the evaluation interval and add duplicate points, shifted by a step of $\Delta = 1e-3$. We generate either $500$ or $333$ datapoints, dependent on whether only the first or also the second derivative is necessary, ensuring that always a total of $1000$ datapoints are created.

We pass the data to the GP resp. LODE-GP and extract the resulting mean function. The first and second derivative are then approximated using forward difference.

To ensure numerical stability in the calculation of the second derivative, we pass the first order forward difference result through a low-pass filter, removing high frequency noise from the result.

To finally calculate the ODE error, we insert the finite difference values for the corresponding derivatives in the original ODEs and average over the error for each ODE.

Table 3: The trained hyperparameters for the bipendulum (top), heating (middle) and three tank (bottom) experiments, for the noisy and noise-free settings. The GP trains three, resp. five, signal variances, for simpler presentation we show the mean of the signal variance parameters. For the LODE-GP three tank system we also use the mean of the two SE kernels that are created.

| | Parameter | Median | Std. deviation | Minimum | Maximum |
|---|---|---|---|---|---|
| LODE-GP noisy | noise | 6.7521e-05 | 0.000149199 | 4.91994e-05 | 0.000470599 |
| | signal variance | 0.0617262 | 0.0351961 | 7.22543e-06 | 0.0879048 |
| | lengthscale | 1.24498 | 3.62295 | 1.18036 | 13.4538 |
| LODE-GP no noise | noise | 0.000306269 | 0.000178327 | 1.7952e-11 | 0.000413706 |
| | signal variance | 0.000298894 | 0.109973 | 6.91032e-06 | 0.354441 |
| | lengthscale | 1.82798 | 0.86266 | 0.69606 | 3.54208 |
| GP noisy | noise | 8.60043e-05 | 1.53343e-05 | 4.62593e-05 | 9.75798e-05 |
| | signal variance | 0.00222937 | 0.00135264 | 0.000539298 | 0.0056802 |
| | lengthscale | 0.786804 | 0.0557031 | 0.709304 | 0.86083 |
| GP no noise | noise | 2.66449e-09 | 2.13689e-08 | 5.74964e-11 | 5.48067e-08 |
| | signal variance | 0.0436976 | 0.0591215 | 3.70462e-05 | 0.152265 |
| | lengthscale | 1.25061 | 0.0776317 | 1.09688 | 1.34608 |
| LODE-GP noisy | noise | 0.000186034 | 3.64772e-05 | 0.000135545 | 0.000243457 |
| | signal variance | 10.0713 | 4.51968 | 0.412769 | 15.3753 |
| | lengthscale | 5.46155 | 0.553268 | 3.7435 | 5.86303 |
| LODE-GP no noise | noise | 1.41396e-06 | 1.97306e-06 | 4.83248e-10 | 5.48139e-06 |
| | signal variance | 6.43588 | 0.520707 | 5.76422 | 7.13915 |
| | lengthscale | 5.06497 | 0.109923 | 4.87297 | 5.2045 |
| GP noisy | noise | 0.000169646 | 2.93415e-05 | 0.000132707 | 0.000218817 |
| | signal variance | 1.26965 | 0.763973 | 0.542204 | 3.03235 |
| | lengthscale | 4.20254 | 0.198991 | 3.8563 | 4.47665 |
| GP no noise | noise | 1.61976e-11 | 5.0974e-13 | 1.57147e-11 | 1.72362e-11 |
| | signal variance | 6.29899 | 1.27418 | 3.36005 | 7.5202 |
| | lengthscale | 5.08887 | 0.175657 | 4.77775 | 5.26459 |
| LODE-GP noisy | noise | 0.00287473 | 0.000523729 | 0.00238355 | 0.00393084 |
| | signal variance | 0.58266 | 0.192348 | 0.323803 | 0.913425 |
| | lengthscale | 5.41985 | 1.00242 | 4.31772 | 7.55746 |
| LODE-GP no noise | noise | 1.18695e-11 | 6.09864e-13 | 1.05086e-11 | 1.23305e-11 |
| | signal variance | 44.2332 | 5.66572 | 32.4225 | 51.8143 |
| | lengthscale | 6.66897 | 0.116321 | 6.42639 | 6.8065 |
| GP noisy | noise | 0.00281939 | 0.000507194 | 0.00176329 | 0.00377263 |
| | signal variance | 0.00104692 | 0.000369078 | 0.000433981 | 0.00150992 |
| | lengthscale | 5.52465 | 1.15024 | 3.01939 | 6.5203 |
| GP no noise | noise | 1.6506e-11 | 3.734e-13 | 1.58496e-11 | 1.72428e-11 |
| | signal variance | 0.00304028 | 0.00140299 | 0.00170012 | 0.006194 |
| | lengthscale | 4.42061 | 0.113491 | 4.20928 | 4.69174 |

Table 4: Median number of eigenvalues greater than zero (i.e. eigenvalue $\lambda > 1\mathrm{e}{-}6$) for the experiments with the LODE-GP and the GP. The total number of eigenvalues for the bipendulum/heating and three tank are 3000 and 5000, respectively.

|  |  | eigenvalues greater zero |
| --- | --- | --- |
| Bipendulum | LODE-GP | 29 |
|  | GP | 76 |
| Heating | LODE-GP | 14 |
|  | GP | 39 |
| Three tank | LODE-GP | 18.5 |
|  | GP | 32.5 |

## B.2 Bipendulum

We illustrate how a LODE-GP covariance function might look like by showing the (1,1) entry for the bipendulum LODE-GP covariance function:

$$\sigma^2 \exp\left(-\frac{(x_1 - x_2)^2}{2\ell^2}\right) \cdot \left(\frac{(x_1 - x_2)^4}{\ell^8} - \frac{6 \cdot (x_1 - x_2)^2}{\ell^6} + \frac{3 + g \cdot (x_1 - x_2)^2}{\ell^4} - \frac{g}{\ell^2} + \frac{g^2}{4}\right)$$

**System details** Since the bipendulum has been thoroughly discussed in the paper, we just briefly mention the system equations, operator matrix and the SNF:

$$\begin{bmatrix} x''(t) + \ell_1 f_1''(t) + g f_1(t) \\ x''(t) + \ell_2 f_2''(t) + g f_2(t) \end{bmatrix} = \mathbf{0} = \underbrace{\begin{bmatrix} \partial_t^2 + \frac{g}{\ell_1} & 0 & -\frac{1}{\ell_1} \\ 0 & \partial_t^2 + \frac{g}{\ell_1} & -\frac{1}{\ell_2} \end{bmatrix}}_{A} \cdot \begin{bmatrix} f_1(t) \\ f_2(t) \\ u(t) \end{bmatrix}$$

case $\ell_1 = 1 \neq \ell_2 = 2$:

$$\underbrace{\begin{bmatrix} 1 & 0 \\ -\frac{1}{2} & 1 \end{bmatrix}}_{U} \underbrace{\begin{bmatrix} \partial_t^2 + g & 0 & -1 \\ 0 & \partial_t^2 + \frac{g}{2} & -\frac{1}{2} \end{bmatrix}}_{A} \underbrace{\begin{bmatrix} 0 & -\frac{4}{g} & \frac{2\partial_t^2 + g}{2} \\ 0 & -\frac{2}{g} & \frac{\partial_t^2 + g}{2} \\ -1 & -\frac{4\partial_t^2 + 4g}{g} & (\partial_t^2 + \frac{g}{2})(\partial_t^2 + g) \end{bmatrix}}_{V} = \underbrace{\begin{bmatrix} 1 & 0 & 0 \\ 0 & 1 & 0 \end{bmatrix}}_{D}$$

To generate the data we use the following solution to the ODE (cf. Figure 7):

$$\begin{bmatrix} f_1(t) \\ f_2(t) \\ u(t) \end{bmatrix} = \begin{bmatrix} -\frac{41 \sin(3\,t)}{100\,(t+1)} - \frac{3 \cos(3\,t)}{5\,(t+1)^2} + \frac{\sin(3\,t)}{5\,(t+1)^3} \\ \frac{81 \sin(3\,t)}{2000\,(t+1)} - \frac{3 \cos(3\,t)}{10\,(t+1)^2} + \frac{\sin(3\,t)}{10\,(t+1)^3} \\ -\frac{3321 \sin(3\,t)}{10000\,(t+1)} + \frac{987 \cos(3\,t)}{500\,(t+1)^2} - \frac{3929 \sin(3\,t)}{500\,(t+1)^3} - \frac{36 \cos(3\,t)}{5\,(t+1)^4} + \frac{12 \sin(3\,t)}{5\,(t+1)^5} \end{bmatrix} \tag{5}$$

**Training details** To train a LODE-GP we generate uniformly spaced training data from Equation 5 in the interval $[1, 6]$ to which we add noise of the form $0.012 \cdot \sigma_n$ for $\sigma_n \sim \mathcal{N}(0, 1)$, i.e. noise with standard deviation 0.012 (2% of the maximal signal value). The data is shown in Figure 7. To validate our results, e.g. using RMSE, we generate uniformly spaced evaluation data from Equation 5 in the interval $[1, 11]$, without adding noise.

## B.3 Heating

**System details** Original system equations:

$$\begin{bmatrix} -f_1'(t) - a \cdot (f_1(t) - f_2(t)) + u(t) \\ -f_2'(t) - b \cdot (f_2(t) - f_1(t)) \end{bmatrix} = \mathbf{0} = \underbrace{\begin{bmatrix} \partial_t + a & -a & -1 \\ -b & \partial_t + b & 0 \end{bmatrix}}_{A} \cdot \begin{bmatrix} f_1(t) \\ f_2(t) \\ u(t) \end{bmatrix}$$

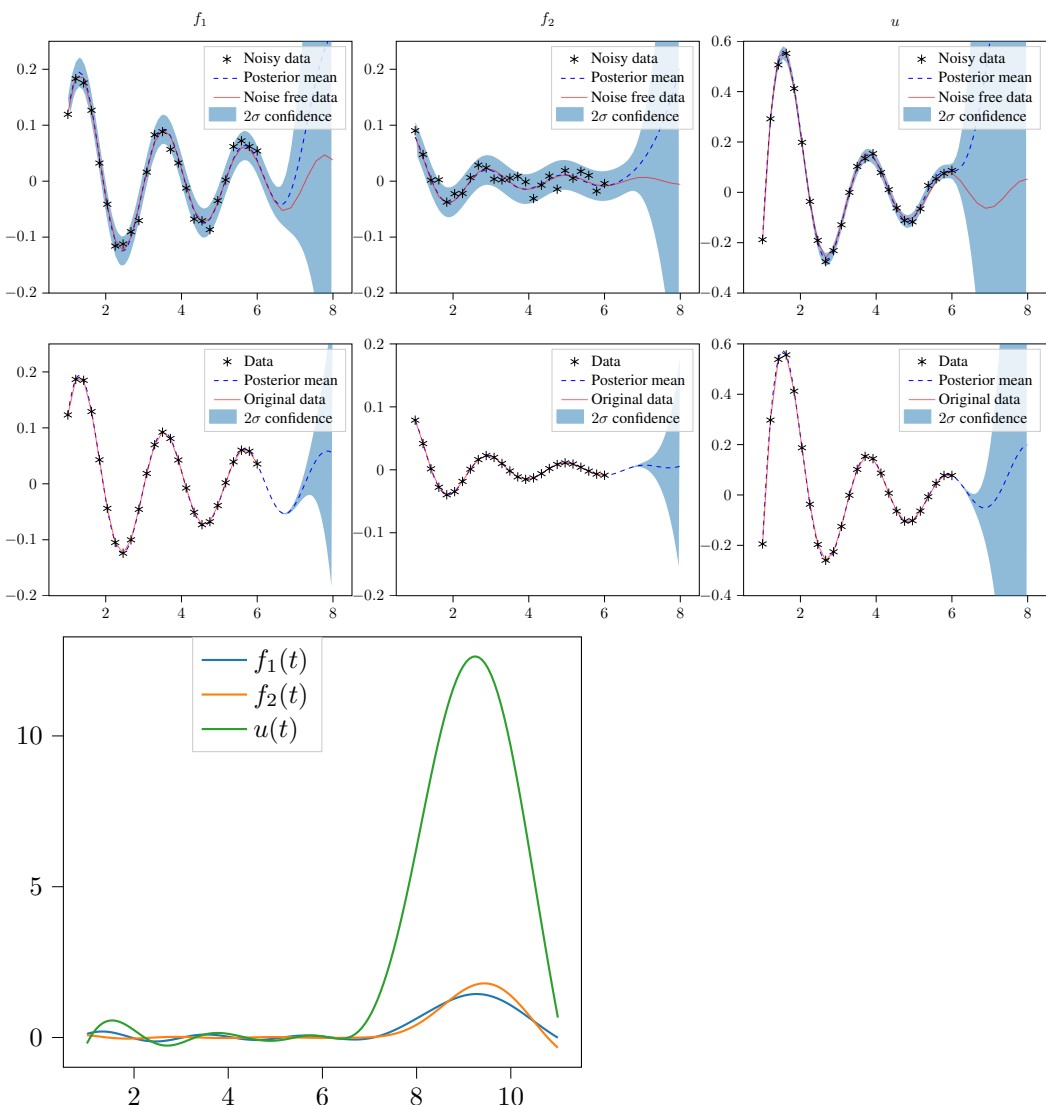

Figure 7: The posterior LODE-GP models for the bipendulum system, trained on noisy data (top) and noise-free data (bottom). The black stars indicate the datapoints (with or without noise), the red line is the solution to the ODEs, the blue dashed line is the LODE-GPs posterior mean, the transparent blue area is the $2\sigma$ confidence interval. In the bottom plot, a sample from the GP is shown.

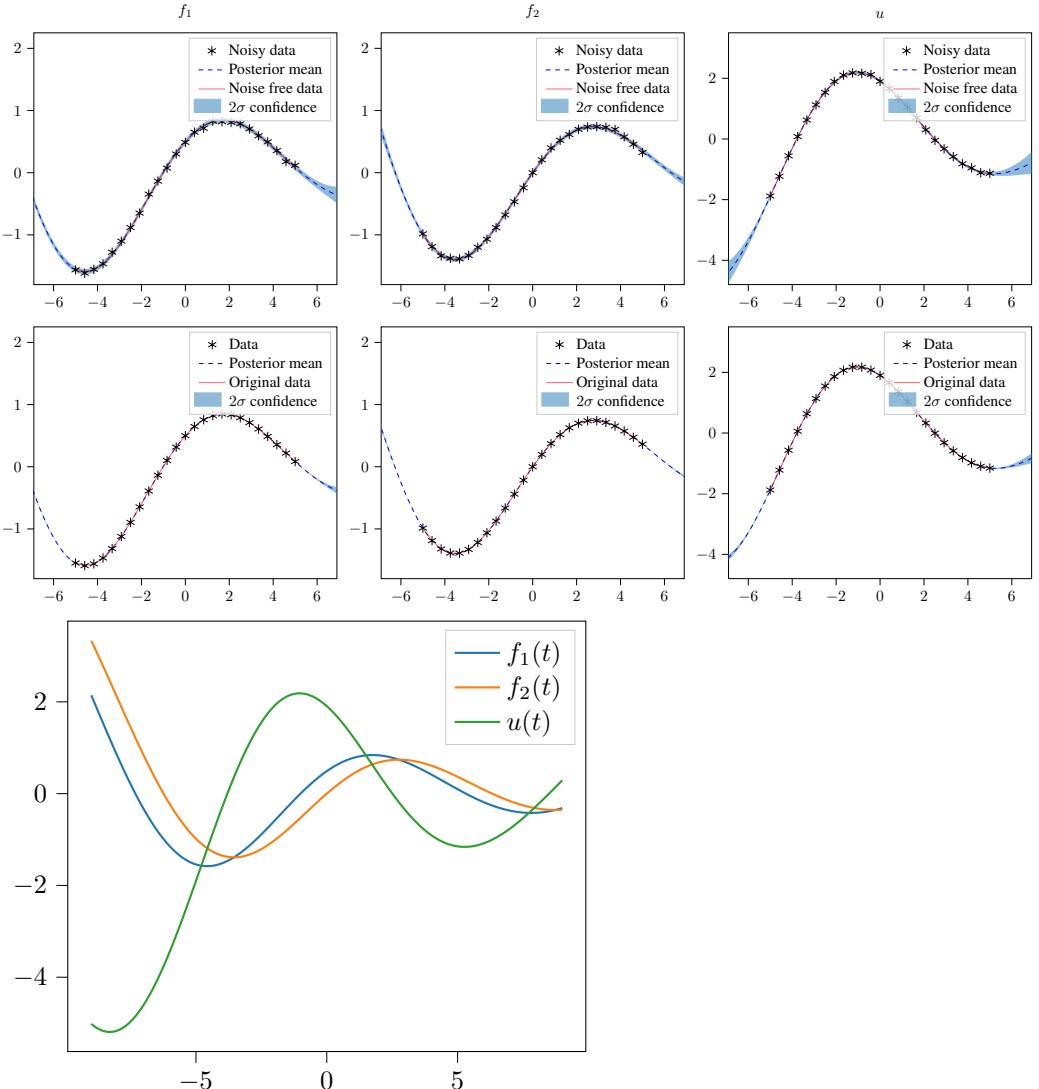

Figure 8: The posterior LODE-GP models for the heating system, trained on noisy data (top) and noise-free data (bottom). The black stars indicate the datapoints (with or without noise), the red line is the solution to the ODEs, the blue dashed line is the LODE-GPs posterior mean, the transparent blue area is the $2\sigma$ confidence interval. In the bottom plot, a sample from the GP is shown.

The SNF of the ODEs with all relevant matrices:

$$\underbrace{\begin{bmatrix} 1 & 0 & 0 \\ 0 & 1 & 0 \end{bmatrix}}_{D} = \underbrace{\begin{bmatrix} 0 & \frac{-1}{b} \\ b & \partial_t + a \end{bmatrix}}_{U} \underbrace{\begin{bmatrix} \partial_t + a & -a & -1 \\ -b & \partial_t + b & 0 \end{bmatrix}}_{A} \underbrace{\begin{bmatrix} 1 & 0 & \partial_t + b \\ 0 & 0 & b \\ 0 & \frac{-1}{b} & \partial_t^2 + (b+a)\,\partial_t \end{bmatrix}}_{V}$$

To generate data, we use the following solution to the heating ODEs (cf. Figure 8).

$$\begin{bmatrix} f_1(t) \\ f_2(t) \\ u(t) \end{bmatrix} = \begin{bmatrix} \frac{1}{2}\cos\left(\frac{1}{2}t\right)\exp\left(-\frac{1}{10}t\right) + \frac{9}{10}\exp\left(-\frac{1}{10}t\right)\sin\left(\frac{1}{2}t\right) \\ \exp\left(-\frac{1}{10}t\right)\sin\left(\frac{1}{2}t\right) \\ \frac{19}{10}\cos\left(\frac{1}{2}t\right)\exp\left(-\frac{1}{10}t\right) - \frac{16}{25}\exp\left(-\frac{1}{10}t\right)\sin\left(\frac{1}{2}t\right) \end{bmatrix} \tag{6}$$

**Training details** To train a LODE-GP we generate uniformly spaced training data from Equation 6 in the interval $[-5, 5]$ to which we add noise of the form $0.02 \cdot \sigma_n$ for $\sigma_n \sim \mathcal{N}(0, 1)$, i.e. noise with

Table 5: The trained parameter values for $a$ and $b$ by the LODE-GP and their relative error from their actual value, for training data with noise (top) and without noise (bottom). Smaller rel. errors are better.

| $a$ | $b$ | relative error $a$ | relative error $b$ |
|---|---|---|---|
| 2.92218 | 0.983648 | 0.0259396 | 0.0163522 |
| 2.9831 | 0.997748 | 0.00563382 | 0.00225164 |
| 2.99933 | 0.991803 | 0.00022441 | 0.00819681 |
| 3.00912 | 1.01786 | 0.00304097 | 0.0178581 |
| 2.92706 | 0.984112 | 0.0243126 | 0.0158881 |
| 2.92016 | 0.968457 | 0.0266129 | 0.0315435 |
| 2.96962 | 0.989062 | 0.0101259 | 0.0109384 |
| 3.00482 | 0.990055 | 0.00160732 | 0.00994511 |
| 3.00571 | 1.00561 | 0.00190434 | 0.00560704 |
| 3.02601 | 0.998836 | 0.0086684 | 0.00116403 |
| 3.00065 | 1.00025 | 0.000217934 | 0.000254375 |
| 3.02079 | 0.989905 | 0.00693057 | 0.0100952 |
| 3.00042 | 1.00132 | 0.000139129 | 0.00132404 |
| 3.00002 | 1.00177 | 0.00000558486 | 0.00176581 |
| 2.97293 | 1.03124 | 0.00902484 | 0.0312367 |
| 2.98821 | 1.01434 | 0.00392941 | 0.0143379 |
| 3.0007 | 0.99991 | 0.00023333 | 0.0000902065 |
| 2.98481 | 1.02509 | 0.00506387 | 0.0250855 |
| 2.99362 | 1.01282 | 0.00212785 | 0.0128176 |
| 3.00045 | 1.00002 | 0.000149713 | 0.0000185651 |

standard deviation $0.02$ (1% of the maximal signal value). The data is shown in Figure 8. To validate our results, e.g. using RMSE, we generate uniformly spaced evaluation data from Equation 6 in the interval $[-9, 9]$, without adding noise.

**Additional results**    We present the exact parameter values trained by the LODE-GP in Table 5. It can be seen that the LODE-GP consistently learn the system parameters $a$ and $b$ with a small relative error, even despite noise.

## B.4 Three tank

Original system equations:

$$
\begin{bmatrix} -f_1'(t) + u_1(t) \\ -f_2'(t) + u_1(t) + u_2(t) \\ -f_3'(t) + u_2(t) \end{bmatrix} = \mathbf{0} = \underbrace{\begin{bmatrix} -\partial_t & 0 & 0 & 1 & 0 \\ 0 & -\partial_t & 0 & 1 & 1 \\ 0 & 0 & -\partial_t & 0 & 1 \end{bmatrix}}_{A} \cdot \begin{bmatrix} f_1(t) \\ f_2(t) \\ f_3(t) \\ u_1(t) \\ u_2(t) \end{bmatrix}
$$

The SNF of the ODEs with all relevant matrices:

$$
\underbrace{\begin{bmatrix} 1 & 0 & 0 \\ -1 & 1 & 0 \\ 1 & -1 & 1 \end{bmatrix}}_{U} \underbrace{\begin{bmatrix} -\partial_t & 0 & 0 & 1 & 0 \\ 0 & -\partial_t & 0 & 1 & 1 \\ 0 & 0 & -\partial_t & 0 & 1 \end{bmatrix}}_{A} \underbrace{\begin{bmatrix} 0 & 0 & 0 & -1 & 0 \\ 0 & 0 & 0 & 0 & -1 \\ 0 & 0 & 1 & 1 & -1 \\ 1 & 0 & 0 & -\partial_t & 0 \\ 0 & 1 & 0 & \partial_t & -\partial_t \end{bmatrix}}_{V} = \underbrace{\begin{bmatrix} 1 & 0 & 0 & 0 & 0 \\ 0 & 1 & 0 & 0 & 0 \\ 0 & 0 & -\partial_t & 0 & 0 \end{bmatrix}}_{D}
$$

We use the following solution to the ODEs (cf. Figure 9).

$$
\begin{bmatrix} f_1(t) \\ f_2(t) \\ f_3(t) \\ u_1(t) \\ u_2(t) \end{bmatrix} = \begin{bmatrix} \exp(-\frac{t}{2}) \\ \exp(-\frac{t}{4}) \\ \exp(-\frac{t}{4}) - \exp(-\frac{t}{2}) \\ -\frac{\exp(-\frac{t}{2})}{2} \\ -\frac{\exp(-\frac{t}{4})}{4} + \frac{\exp(-\frac{t}{2})}{2} \end{bmatrix} \tag{7}
$$

**Training details**  To train a LODE-GP we generate uniformly spaced training data from Equation 7 in the interval $[1, 6]$ to which we add noise of the form $0.08 \cdot \sigma_n$ for $\sigma_n \sim \mathcal{N}(0, 1)$, i.e. noise with standard deviation $0.08$ (10% of the maximal signal value). The data is shown in Figure 9. To validate our results, e.g. using RMSE, we generate uniformly spaced evaluation data from Equation 7 in the interval $[1, 11]$, without adding noise.

For the three tank system we hard coded the case $-x = x$, which is a legal operation since we can take out the $-1$ as part of a left matrix multiplication and integrate it in the $U$ matrix, which does not influence the LODE-GP.

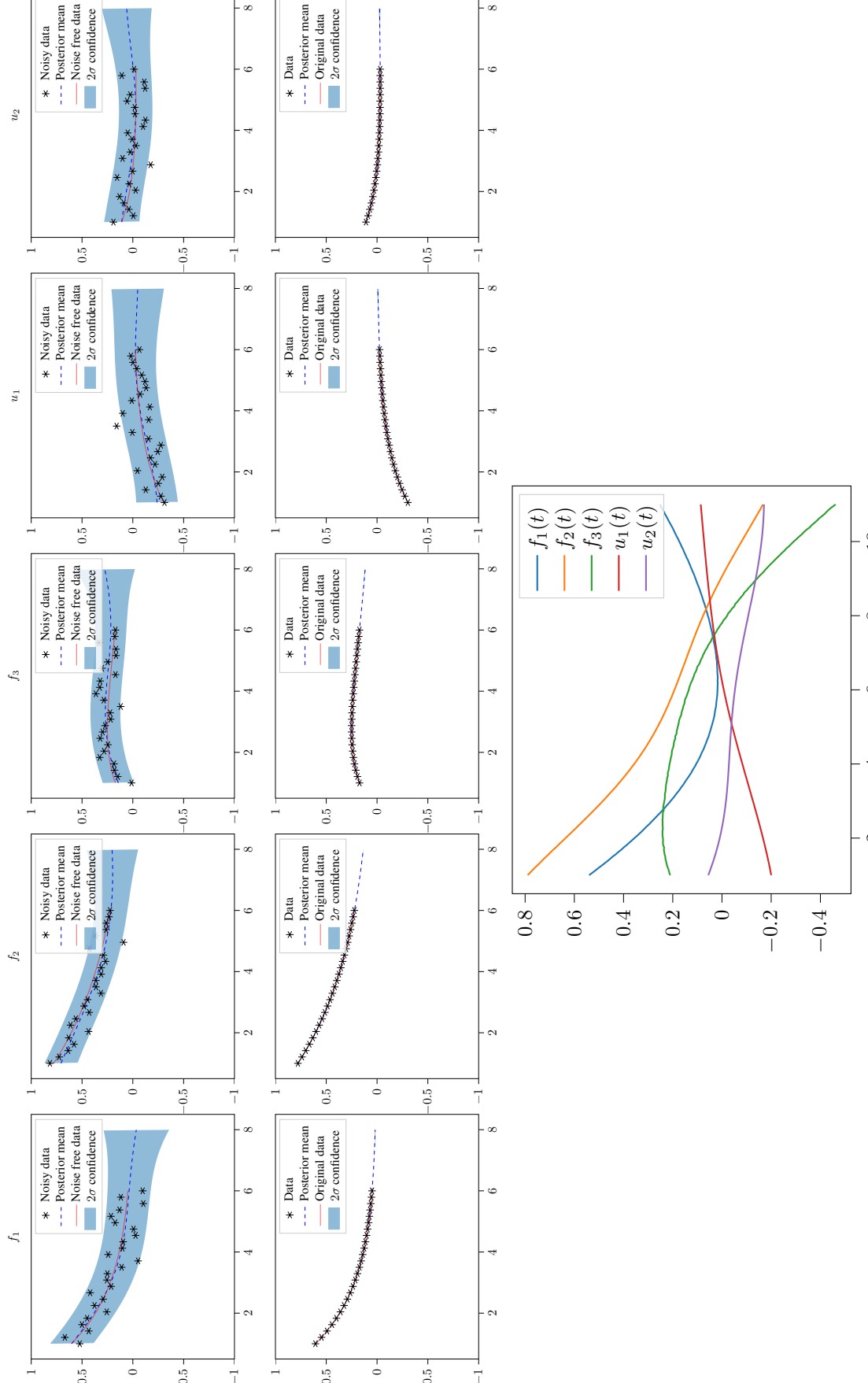

Figure 9: The posterior LODE-GP models for the three tank system, trained on noisy data (top) and noise-free data (bottom). The black stars indicate the datapoints (with or without noise), the red line is the solution to the ODEs, the blue dashed line is the LODE-GPs posterior mean, the transparent blue area is the $2\sigma$ confidence interval. In the bottom plot, a sample from the GP is shown.

**Comparison to Latent Force Model**   We compare our LODE-GP with the LFM introduced by [2]. To do so we set the variables from Equation 3 in [2] as follows: $D_q = 0$, $B_q = 0$, $S_{rq} = \begin{bmatrix} -1 & 0 \\ -1 & -1 \\ 0 & -1 \end{bmatrix}$. Further we calculate the covariance function as the solution of the integral $k = \int_0^{t_2} \int_0^{t_1} \exp^{-(x_1 - x_2)^2} dx_1 dx_2$, effectively setting $\ell = 1$ and $\sigma = 1$. Following the steps of [2], we get a GP that can estimate the solution of the three tank systems differential equations.

The resulting GP marginalizes the (in their model considered) latent function $u_1$ and $u_2$ and only considered the three data channels $f_1$, $f_2$, and $f_3$. To calculate the ODE error, we inserted the original data for the channels $u_1$ and $u_2$ into the calculation.

For a fair comparison with our LODE-GP, we also marginalize $u_1$ and $u_2$ there. Similarly as for the LFM, we have set $\ell = 1$ and $\sigma = 1$.

The resulting ODE errors of the two models are as shown in Table 6. The performance of the marginalized LODE-GP is better but comparable to the LFM. The performance of the full LODE-GP, having learned all 5 channels and also set all lengthscales and signal variances to 1, shows a signifcant increase in performance.

Table 6: The ODE error of the LFM, the small LODE-GP, and the full LODE-GP. Smaller is better.

|  | ODE 1 error | ODE 2 error | ODE 3 error |
| --- | --- | --- | --- |
| (marginalized) LFM | 0.042816 | 0.507017 | 0.013084 |
| marginalized LODE-GP | 0.031331 | 0.025065 | 0.008595 |
| full LODE-GP | 1.210e-05 | 1.285e-05 | 1.081e-05 |

## C   Discussion of the code and runtime

We use GPyTorch as the core of our code and build our LODE-GP on top of their work, which makes it an essential part of our implementation. Nevertheless, it could be replaced with other GP libraries if the kernel would be rewritten for those. We use SageMath to calculate the symbolic SNF, given the system of ODEs. In our current implementation, SageMath is a necessary library, but could be replaced by any other computer algebra system, with a Python interface, that symbolically solves the SNF for matrices $A \in \mathbb{R}[x]^{m \times n}$ over the polynomial ring $\mathbb{R}[x]$, and additionally the function field $\mathbb{R}(a_1, \ldots, a_k)[x]$ if there are parameters in the ODEs.

Training a LODE-GP on 25 multidimensional[5] datapoints with the stated training specifications needed, in the median, between 3 and 5 seconds per training and, 7 seconds for the three tank system, due to its greater size, as shown in Table 7. This result is in accordance to our findings in Section 5, where we discuss that the $\mathcal{O}(n^2)$ calculation takes longer, whereas the $\mathcal{O}(n^3)$ stays roughly the same. Due to the small number of datapoints, the $\mathcal{O}(n^2)$ calculations have a strong influence on the runtime.

The GP needed roughly 15 seconds per training run.

Additionally, generating the model, i.e. the SNF calculation, preparing the differentiated covariance function and initializing the model object, took averagely 0.3 seconds. Of this time, the SNF calculation takes, on average, 0.15 seconds. These calculations are performed only once for the model and are neglegtable, compared to the training runtime.

During development, we encountered an interesting problem we want to document for other developers. Namely that PyTorch only trains parameters that are part of the model object itself (i.e. which can be called using the `self` keyword). For example, we can't just use a list, that is part of the model object, in which we store the covariance functions, this would cause PyTorch to not recognize the parameters and not train them. In our case, the various kernels and ODE parameters, that are added dynamically, required us to give each parameter and kernel a name and add it to the kernel individually. We also store a position matrix with references to the respective kernel at that position, e.g. for

---

[5]The data is three dimensional for the bipendulum and heating system and five dimensional for the three tank system.

Table 7: Median training runtime, in seconds, for 300 iterations on 25 datapoints, for the LODE-GP and the GP, for each of the systems. Smaller is better.

|  |  | training runtime |
| --- | --- | --- |
| Bipendulum | LODE-GP | 5.02 |
|  | GP | 2.34 |
| Heating | LODE-GP | 3.12 |
|  | GP | 2.2 |
| Three tank | LODE-GP | 7.49 |
|  | GP | 1.07 |

the bipendulum covariance function we have a $3 \times 3$ matrix with different kernel objects for each position.

Additionally, we want to highlight the On-Line Encyclopedia of Integer Sequences, which we used to find the construction formula of the number sequence for the general SE kernel derivative.[6]

## D   Proof for base covariance function construction

We prove the following Lemma

**Lemma 1.** *Covariance functions for solutions of the scalar linear differential equations $d \cdot f = 0$ with constant coefficients, i.e. $d \in \mathbb{R}[\partial_t]$, are given by Table 8 for $d$ is primary, i.e. a power of an irreducible real polynomial. In the case of a non-primary $d$, each primary factor $d_i$ of $d = \prod_{i=0}^{\ell-1} d_i$ is first translated to its respective covariance function $k_i$ separately before they are added up to give the full covariance function $k(t_1, t_2) = \sum_{i=0}^{\ell-1} k_i(t_1, t_2)$.*

Table 8: Primary operators $d$ and their corresponding covariance function $k(t_1, t_2)$.

| $d$ | $k(t_1, t_2)$ |
| --- | --- |
| $1$ | $0$ |
| $(\partial_t - a)^j$ | $\left( \sum_{i=0}^{j-1} t_1^i t_2^i \right) \cdot \exp(a \cdot (t_1 + t_2))$ |
| $((\partial_t - a - ib)(\partial_t - a + ib))^j$ | $\left( \sum_{i=0}^{j-1} t_1^i t_2^i \right) \cdot \exp(a \cdot (t_1 + t_2)) \cdot \cos(b \cdot (t_1 - t_2))$ |
| mid $0$ | $\exp(-\frac{1}{2}(t_1 - t_2)^2)$ |

*Proof.* For a function $f$ we have $1 \cdot f = 0$ if and only if $f = 0$. Such functions are described by the zero covariance function (and zero mean function).

The real differential equation $(\partial_t - a)^j \cdot f = 0$ only has the analytic solutions given symbolically by $\sum_{i=0}^{j-1} a_i \cdot t^i \cdot \exp(a \cdot t)$ for arbitrary $a_i \in \mathbb{R}$. This is a finite dimensional space and hence (as for linear regressions problems) a covariance function is given by $\sum_{i=0}^{j} \left( t_1^i \cdot \exp(a \cdot t_1) \right) \cdot \left( t_2^i \cdot \exp(a \cdot t_2) \right) = \left( \sum_{i=0}^{j-1} t_1^i t_2^i \right) \cdot \exp(a \cdot (t_1 + t_2))$.

The real differential equation $((\partial_t - a)^2 + b^2)^j = ((\partial_t - a - ib)(\partial_t - a + ib))^j$ only has the analytic solutions given symbolically by $\exp(a \cdot t) \cdot \sum_{i=0}^{j-1} t^i \cdot (a_i \cdot \cos(b \cdot t) + b_i \cdot \sin(b \cdot t))$ for arbitrary

---

[6] http://oeis.org/A096713

$a_i \in \mathbb{R}$. This is again finite dimensional space and hence a covariance function is given by

$$
\sum_{i=0}^{j} \left( t_1^i \cdot \exp(a \cdot t_1) \cdot \cos(b \cdot t_1) \right) \cdot \left( t_2^i \cdot \exp(a \cdot t_2) \cdot \cos(b \cdot t_2) \right)
$$

$$
+ \sum_{i=0}^{j} \left( t_1^i \cdot \exp(a \cdot t_1) \cdot \sin(b \cdot t_1) \right) \cdot \left( t_2^i \cdot \exp(a \cdot t_2) \cdot \sin(b \cdot t_2) \right)
$$

$$
= \left( \sum_{i=0}^{j-1} t_1^i t_2^i \right) \cdot \exp(a \cdot (t_1 + t_2)) \cdot (\cos(b \cdot t_1) \cdot \cos(b \cdot t_2) + \sin(b \cdot t_1) \cdot \sin(b \cdot t_2))
$$

$$
= \left( \sum_{i=0}^{j-1} t_1^i t_2^i \right) \cdot \exp(a \cdot (t_1 + t_2)) \cdot \cos(b \cdot (t_1 - t_2)).
$$

The factor $\cos(b \cdot (t_1 - t_2))$ is sometimes called the cosine covariance function.

For a smooth function $f \in C^\infty(\mathbb{R}, \mathbb{R})$, the equation $0 \cdot f = 0$ poses no restriction. Hence, by [40, Prop. 1], the squared exponential covariance function yields a GP with realizations densely contained in the space $C^\infty(\mathbb{R}, \mathbb{R})$. □

## E  Licenses and versions

We use Python 3.9.7, which uses the PSF license agreement.

We use GPyTorch version 1.5.0 for our experiments, GPyTorch is distributed under the MIT license.

We use SageMath version 9.5, released 2022-01-30, for our experiments, SageMath is distributed under the CC BY-SA 4.0 license.