# OpenReview forum: "Constraining Gaussian Processes to Systems of Linear Ordinary Differential Equations"
_NeurIPS.cc/2022/Conference — NeurIPS 2022 Accept_

### Official Review · Reviewer_6STF · 2022-06-23

**Rating:** 6
**Confidence:** 4
**Soundness:** 3 good
**Presentation:** 3 good
**Contribution:** 2 fair

**Summary:**

This paper proposes a method, based on Smith normal form, to construct Gaussian processes indexed by R such that they satisfy a specified differential equation.



**Questions:**



Q1 - I guess \ell_1 should really be \ell_2 in entry (3,2) of A in eq. (1) ?

**Limitations:**

Its fine.

**Strengths And Weaknesses:**


S1 - The paper is clearly written.

W1 - While the proof of Lemma 1 is indeed trivial enough to defer to appendix, it may be useful to point out in the main text that (a) the proof is indeed given in appendix, and (b) the covariances are obtained by expanding f in a basis of the null-space of d with i.i.d standard normal coefficients.

W2 - The entire noise free data is not shown in fig 2. It woudl certainly be interesting to see how well the method extrapolates.

---

> ### Author Response · Authors · 2022-08-02
> **Response to Reviewer 6STF**
>
> Thank you very much for your review and your input to improve our paper.
>
> W1: We have added the reference to the proof for Lemma 1 and have quickly discussed its idea in the paper as well.
>
> W2: Thanks to your comment we have now included the full noise-free data in Figure 2.
>
> Q1: You are right, thank you for spotting this mistake! We have, of course corrected this, thanks to you.
>
> Again, we want to thank you for your comments.

---

### Official Review · Reviewer_JWZD · 2022-07-08

**Rating:** 6
**Confidence:** 3
**Soundness:** 3 good
**Presentation:** 3 good
**Contribution:** 3 good

**Summary:**

The authors introduce a new method which tackles the important and much discussed problem of learning the solutions to systems of ordinary differential equations from noisy data. The authors construct a Gaussian process prior for the solutions to the equation, which has been a common approach in previous work in the area. In particular the authors attempt to solve the problem in the case of linear differential operators for systems which are underspecified and uncontrollable. The proposed method relies on the symbolic calculation of the Smith normal form of the differential operator representing the system, which the authors use to construct a covariance over both the system outputs and inputs, ensuring that samples obey the differential equation in question. The proposed method is validated on 3 synthetically constructed physical systems, providing convincing evidence that the method works.


**Questions:**

Below are my questions/suggestions for the authors, which are mostly a summary of what is discussed in the weaknesses section. I've numbered the points separately to allow the authors to respond more easily.

1. Please can the authors comment on the significance of the ability to model noncontrollable systems, and provide some real world motivation? In doing so could the authors expand the discussion of how this work represents a significant contribution over [34]?
2. How does the proposed method compare to existing models of this type from problems that are fully specified?
3. Can the authors comment on the relationship between this model and LFMs? If they are indeed equivalent, the authors should discuss this in the text, and add a comparison for system 3 if possible?
4. I suggest that the authors add standard errors/deviations to table 2, and indicate if the standard errors are narrow enough to indicate one model being significantly better than the other.

If the above points are addressed I would be more than happy to raise my score.


Below are some very minor typos/formatting/wording issues:

* Line 23 "allow to"
* Line 193 not sure what "All experiments ran numerically stable and robust using Adam [28]." means, please rephrase.
* Line 233 "The LODE-GP can be constructed using parameters." this is not clear, please rephrase.
* Many of the figures don't have titles/axis labels, so are difficult to read, although mostly the relevant information is in the caption/legend. For figure 2 particularly it is not at all clear which plot is which output, presumably it is f1, f2, u from left to right, but please clarify via titles.

**Limitations:**

I do not believe there are any ethical concerns with this work or problems with potential negative societal impact.

Aside from what I have already mentioned, I believe the authors reasonably fairly represent the limitations of their work, although I do think the systems shown in the evaluation are somewhat similar, and it would be good to get some idea of how these models would perform on real world data, not just synthetic systems.

**Strengths And Weaknesses:**

Strengths
---------

Overall I enjoyed reading the paper, and think it has a number of strengths. The paper for the most part is very clearly written, and introduces the problem, and the pre existing solutions, well. I particularly like the use of explicit examples at key points in the exposition which help the reader to understand how the proposed method is applied in practice. The figures are mostly clear and useful.

The idea presented in the paper is quite an elegant one in my opinion, and seems to be a sensible way to tackle the problem. The fact that the method can be used to construct priors which consist of exact solutions to the given system is compelling.

The method seems to work on a number of different problems, and in some cases shows good performance relative to standard GPs, although I do have some concerns about the evaluation, as listed below.

I am not expert in the theoretical tools such as the Smith normal form used in the paper, and so I can not vouch for the correctness of those parts with 100% certainty. That being said, the results given in the paper are correct to the best of my knowledge.

Weaknesses
----------

There are a few weaknesses with the paper, mostly related to the evaluation of, and motivation for, the proposed methods.

**Motivation for uncontrollability**
The authors say that their methods is an improvement on [34] because it can deal with uncontrollable systems.  As the authors note, in 2 of the experiments which are controllable, the proposed method generates exactly the same results as the previous work of [34]. To me, there is not sufficient discussion of the reasons that the uncontrollable case is an important problem, and thus why the proposed method represents an important contribution. I am not an expert in control theory, so it may be that I am unaware, but I think it would be good to motivate this problem more clearly in the text, and explain why it is important to be able to model such systems. Additionally for the experiment where the proposed model is novel (three tanks), the performance seems to be worse than a vanilla GP in terms of it's representation of the data.

**Comparison with related work**
Presumably the proposed method can work for problems that are well specified, as well as those discussed in the text that are underspecified. Currently there is no indication of how this method performs in that case, and how well it performs relative to other prior work. I think it would improve the paper considerably to include some comparisons.

The authors claim that "No other state of the art in probabilistic ML exists for our class of differential equations" I am not entirely sure that this is correct. I believe that is is possible to solve some of the problems using the framework of latent force models (LFMs) [Alvarez et al. 2009] which represent the input functions as GPs, and use the Green function of the differential operator to map to exact outputs. Most work on LFMs deals ODE systems which are not coupled, i.e. the part of $A$ relevant to the outputs is diagonal, so it may be additional effort to get systems 1 & 2 running in the LFM framework, but I believe system 3 can easily be recast in terms of LFMs. System 3 is equivalent to the model in equation 3 of [Alvarez et al. 2009], when $D_q = 0$, $B_q=0$ and $S_{rq}=\\begin{pmatrix}
    -1 & 0\\\\
    -1 & -1 \\\\
    0 & -1
\end{pmatrix}$, the expressions given in later equations can then be used to compute the covariance of the outputs. It would be great to have a comparison between the proposed model and LFMs for system 3, at least i think a comment on this equivalence is necessary. Again I may be mistaken in this point, but I would appreciate a comment from the authors if so.

**Evaluation**
A minor point, but I'm not sure "ODE error" shown table is interesting to compare with the GP. The metric is useful as a check that the model is working, but it doesn't seem fair to compare with the GP since the information about the differential equation is not available to the GP, and the error should be small for the proposed model almost by definition.

Table 2 shows results for 10 repeats, with bold text indicating best performance, but does not include the standard deviation or error on these results. This means it is not possible for the reader to assess the significance of the results. Please add sandard errors or deviations to this table

##### References
- [Alvarez et al. 2009] Alvarez, Mauricio, David Luengo, and Neil D. Lawrence. "Latent force models." Artificial Intelligence and Statistics. PMLR, 2009.

---

> ### Author Response · Authors · 2022-08-02
> **Main Response to Reviewer JWZD**
>
> We thank you very much for your comprehensive review and the great questions.
> Thank you particularly for "enjoyed reading the paper". :-)
>
> Below we discuss some of your questions with additional discussions in the other comments.
>
> > Additionally for the experiment where the proposed model is novel (three tanks), the performance seems to be worse than a vanilla GP in terms of it's representation of the data.
>
> Thanks to your comment regarding standard deviations, we have redone the experiments with 20 iterations per experiments (see below).
> We do not argue that LODE-GP perform better than GPs in RMSE, since SE-GPs more flexibly fit any data.
> Still, the ODE error favours the LODE-GP, since they yield a regression model following the physical interpretation.
> Of course, this is obvious, since the LODE-GP satisfies the ODE by definition.
>
> The only instance where the model performances are not really comparable is the evaluation RMSE for the Bipendulum experiment, where we already discussed in the paper that the physical correctness of the LODE-GP caused the massive spike, shown in Figure 4, to get back to the zero mean prior as fast as possible.
>
>
> |            |         | training RMSE     | evaluation RMSE   | loss               | mean ODE error            |
> | ---------- | ------- | ----------------- | ----------------- | ------------------ | ------------------------- |
> | Bipendulum | LODE-GP | 0.060 $\pm$ 0.060 | 0.106 $\pm$ 0.258 | -1.638 $\pm$ 0.567 | **5.135e-07** $\pm$ 1.607e-06 |
> |            | GP      | **0.008** $\pm$ 0.030 | **0.044** $\pm$ 0.017 | **-2.072** $\pm$ 0.215 | 0.064 $\pm$ 0.023         |
> | Heating    | LODE-GP | **0.006** $\pm$ 0.001 | **0.170** $\pm$ 0.068 | **-2.089** $\pm$ 0.100 | **0.008** $\pm$ 0.007         |
> |            | GP      | 0.010 $\pm$ 0.001 | 0.333 $\pm$ 0.055 | -1.628 $\pm$ 0.051 | 0.218 $\pm$ 0.070         |
> | Three tank | LODE-GP | **0.023** $\pm$ 0.004 | 0.078 $\pm$ 0.046 | -0.949 $\pm$ 0.063 | **1.360e-05** $\pm$ 4.919e-06 |
> |            | GP      | 0.028 $\pm$ 0.005 | **0.058** $\pm$ 0.031 | **-0.974** $\pm$ 0.071 | 0.040 $\pm$ 0.020         |
>
>
>
> **Comparison to related work in the underspecified case**
>
> If the system is not underspecified ("underspecified" defined as "the solution space is infinite dimensional"), then the solution space is a finite dimensional vector space. Then, our approach boils down to Bayesian linear regression in the solution space.
>
>
> **Additional discussions**
> Below you will find additional comments that provide:
> (1) An extended discussion about controllability and the relevance of uncontrollable systems.
> (2) A discussion about the comparison of the LODE-GP to the LFM and its application on the three tank system.
>
>
> We also tackled your minor comments and will also fix the figures.
>
> We again want to thank you for your questions, which we hope to have answered here.

---

> > ### Comment · Reviewer_JWZD · 2022-08-04
> > **Response**
> >
> > Thanks for this response, it has help clarify a number of points for me.
> >
> > I have raised my score by 1 point to 6 (and increased contribution score to 3), in light of the responses. I have mention a couple of things in the responses to the specific rebuttals, I would appreciate if you could add these in the camera ready, thanks.

---

> > > ### Author Response · Authors · 2022-08-05
> > > **Thank you**
> > >
> > > We thank you very much for your feedback and the new score.
> > >
> > > We will take care to implement your comments into the camera ready version of the paper!

---

> ### Author Response · Authors · 2022-08-02
> **Motivating controllability**
>
>
> **Motivating Controllability**
>
> (For systems as considered in our paper:)
> The solution space decomposes into a direct sum (for PDEs we would need to deal with filtrations instead) of two summands. The first summand is the controllable part and corresponds to the zeros on the diagonal of the Smith form. It is isomorphic (as a module over the ring of linear ordinary differential operators with constant coefficients $\mathbb{R}[\partial]$) to a function space of freely choosable functions. It is called controllable, because "you can do anything with it". The second summand is the uncontrollable part and corresponds to the non-zeros on the diagonal of the smith form. It is isomorphic to finite dimensional space, e.g. $\mathbb{R}\exp(t)+\mathbb{R}\exp(-2t)$. It is called non-controllable, because finitely many initial conditions determine the solution set. One says a system is controllable, if the second summand is zero. (For PDEs, everything is much more complicated.) All of this does not necessarily have anything to do with control theory.
>
> Systems from typical undergraduate courses usually only have an uncontrollable part, e.g. $f''+af'+bf=0$ or $f'=Af$ for a square real matrix $A$ are guaranteed to be of this form. Systems in engineering often only have a controllable part. (Let us not get into details here, what controllers do to such systems.) Random systems (big enough, differential equations with random terms and random coefficients, more functions then equations) usually contain both controllable and uncontrollable parts. Hence, both cases appear, and they appear jointly.
>
> We are not sure how well-known this is. Feel free to indicate in the discussion phase, if it makes sense to include this or parts of it in the paper or its appendix. Or whether we should rephrase this using more involved mathematical theories like jet bundles, or Hilbert series and Kolchin's dimension polynomial.
> Otherwise, we could also could discuss how this all fits in control theory.

---

> > ### Comment · Reviewer_JWZD · 2022-08-04
> > **Response**
> >
> > I think the above discussion is interesting and, in my opinion, not common knowledge for machine learning researchers (although I may be biased here due to my background). In addition to the above discussion which is quite technical/mathematical I would appreciate if the authors could add a line or so on the practical necessity of modeling uncontrollability (i.e some real world applications), perhaps from control or some other area.

---

> ### Author Response · Authors · 2022-08-02
> **Comparison to Alvarez et al. and experiments with the three tank systems**
>
>
> **Comparison to Alvarez et al. [Al09]**
>
> [Al09] constructs (different, but equivalent in the sense of dominating each other) priors for a subset of our class of ODEs.
> - For the trivial example $f'=g$, [Al09] takes a covariance function $k_f$ and constructs $k_g=\int\int k_f$. LODE-GPs take a coavariance function $k_g$ and construct $k_f=\partial\partial k_g$.
> - [Al09] uses pushforwards by linear integral operators with variable coefficients; this makes symbolic integration of covariance functions necessary. LODE-GPs uses pushforwards by linear differential operators with constant coefficients, easily applied to typical covariances functions.
> - [Al09] uses pushforwards by actually solving the differential equations; this is easy for decoupled systems of 1st and 2nd order, harder for higher order, and much harder for coupled systems. LODE-GPs do not solve the system, but rewrite it differently.
> - [Al09] needs (i) ODEs (ii) restricted to first or second order, which are (iii) decoupled (i.e. the $y_q$'s for different $q$ in equations (3) or (4) do not interact). LODE-GPs only work for ODEs, hence are much more universally applicable.
> - [AL09] only allows data from and predictions of the "observable" functions, whereas our covariance functions flexibly allow data from and predictions of all ("observable" and "unobservable") functions.
>
>
> **Comparing the three tank example to [Al09]**
>
> And yes, the three tank example can be phrased in the LFM framework, as you claim.
>
> We have implemented a version of Alvarez et al. manually (we didn't find a viable version on Github or referenced in the paper).
> By setting $D_q = 0, B_q = 0$ and $S_{rq}$, as you suggested.
>
> For simplicity we have used the squared exponential covariance on the diagonal of the base kernel with $\ell = 1$ and $\sigma=1$ and solved:
> $k = \int^{t_2}_0 \int^{t_1}_0 exp^{-(x_1 - x_2)^2} dx_1dx_2$
>
> The resulting covariance function $K = k\cdot I_2$  is now contained in the GP corresponding to the integral on the right hand side of the equation $L_{rq}... = \dots$ on page 11.
> When LODE-GPs are restricted (as the LFM) to the first three functions, the covariance function is the pushforward via $S_{rq}$ (as above) of two independent SE covariances.
> Hence, in this special case of LODE-GPs, both methods are mostly equivalent, only that the base covariance is different. (Of course, both methods allow different base covariances anyway, hence you could view both methods being equivalent.)
>
> Keeping the parameters fixed and just doing inference on the evaluation datapoints results in the ODE errors:
> [0.042816, 0.507017, 0.013084]
>
> If we constrict our LODE-GP as described above, we get the following ODE errors:
> [0.031331,  0.025065,  0.008595]
>
> (These two sets of errors are significantly bigger than zero, as the unobservable functions are marginalized out.)
>
> These values are slightly better than our LFM implementation, therefore we would argue that the LODE-GP performance and the LFM are (at least) comparable in representing the system and might even favour the LODE-GP.
>
> For reference, our LODEGP for the full three tank system, with also manually setting all variances and lengthscales to $1$ returns the following ODE errors:
> [1.210e-05,  1.285e-05,  1.081e-05]
>
> We hope this comparison is considered fair and valid by you.
> Since we can't guarantee that the code is 100% correct, we take this result with a tiny grain of salt and would be interested in trying it out with a tested code, which is able to deal with the setting $D_q = 0, B_q = 0$ without dividing by zero.
> If you know where a running version exists, or you have one at your disposal, we would be happy to try it out.
>
> Further, if you consider this comparison important to mention in the paper, please say so and we will add it either to the paper or the appendix, depending on the available space.

---

> > ### Comment · Reviewer_JWZD · 2022-08-04
> > **Response and requests for CR**
> >
> > Thanks for this very comprehensive response! I think this fully addresses my concern and has improved my view of the contribution of the work. I think the above discussion would be interesting to include in the paper in some form, perhaps mostly in the appendix, although I do think two things should be changed in the camera ready:
> >
> > 1) Add a line or two in the related work describing these differences.
> > 2) Change line 189/190 to reflect this discussion, I think it is still somewhat misleading.

---

### Official Review · Reviewer_Wp1C · 2022-07-09

**Rating:** 6
**Confidence:** 4
**Soundness:** 3 good
**Presentation:** 3 good
**Contribution:** 3 good

**Summary:**

The paper describes how to constrain realisations of a Gaussian process $f \sim GP(m, k)$ to satisfy $A f = 0$, where $A$ is a linear, matrix-valued differential operator with constant coefficients.
This is done by computing the Smith normal form of $A$, which is a (symbolic) decomposition of $A$ into
$U D V = A$, or equivalently $D = U^{-1} A V^{-1}$, and identifying the null space of $D$.

This is most closely related to the work by Hegermann (2018) and Hegermann (2021), who tackle a similar problem using a Gröbner basis.
The difference between the submitted work and the aforementioned papers is the use of a Smith normal form, which allows for non-controllable systems, and can be implemented with standard software. The Smith normal form also restricts the class of problems that can be tackled to linear, homogenous, ordinary differential equations with constant coefficients.

**Questions:**


* Abstract: what does "novel algorithmic and symbolic construction" mean? It is written frequently, but I don't understand "algorithmic and symbolic construction".
* Line 87: Please explain the following notation in the paper $\mathbb{R}_\preceq$. I suppose it refers to some sort of positive-definiteness?
* Line 223f: "We additionally analyse the training runtime and see that...": are those results in some figure/table? I can't seem to find the results


**Limitations:**

It would be useful if the limitations of the method would be discussed more centrally. What do I mean by this? Line 68 states "but are restricted to ODEs". Lines 72f. describe the limitations as "restricted to linear ODEs". Line 140 states: "Consider a system of linear homogenous ODEs with constant coefficients". Line 172 says: "This is generally impossible for PDEs". Line 255 states: "follow a given system of linear homogeneous ODEs".
It feels like a reader has to search the entire document to learn what the limitations of the method are.


**Strengths And Weaknesses:**

## Strengths
Overall, the methodology seems to be sound, the related work seems to be discussed adequately, and the experiments support the claims.
I would expect that the method described in the paper will be useful to some people. The paper is at the interface of differential equation models and Gaussian processes, which seems to be a relevant subject for Neurips.


## Weaknesses
* One criticism of the method is that the algorithm is restricted to linear, homogenous ODEs with constant coefficients and that there may not be much hope to generalise to nonlinear/nonhomogenous problems or even partial differential equations.
* From a didactical viewpoint, I would appreciate it if the paper could provide more intuition for Lemma 1. Maybe a few equations would help.
* This is minor, but I would like to see samples from the constrained Gaussian process, perhaps in one of Figures 2, 4, 5, or 6?
* Why does only Figure 2 show the credibility intervals? The remaining figures only discuss the mean. Is there a specific reason?

---

> ### Author Response · Authors · 2022-08-02
> **Response to Reviewer Wp1C**
>
> Thank you very much for your review, your comments, and your input to improve our paper. We hope to answer your open questions below.
>
> >One criticism of the method is that the algorithm is restricted to linear, homogenous ODEs with constant coefficients and that there may not be much hope to generalise to nonlinear/nonhomogenous problems or even partial differential equations.
>
> You are correct that this work is limited to linear ODEs with constant coefficients. However, it is not limited to initial value problems or any other special forms (semi-linear, explicit, decoupled, ...) of ODEs, as is often the case in the literature.
> The limitation to homogenous problems is actually not present due to the fundamental theorem on homomorphisms: assuming a linear non-homogenous ODE with constant coefficients $A\cdot\mathbf{f}=\mathbf{y}$, then any solution is of the form $\mathbf{f}=\mathbf{f}_0+\mathbf{f}_p$, where $f_0$ is a solution of $A\cdot\mathbf{f}=0$ and $\mathbf{f}_p$ is a particular solution. Then, any solution can be approximated by the Gaussian process $GP(\mathbf{f}_p, k)$, where $k$ is the covariance function constructed in our paper.
> Of course, the presented approach can not deal with nonlinear ODEs or PDEs. This necessitates cited works like [34] and [47,48,49], which also only work under their respective assumptions.
>
> >From a didactical viewpoint, I would appreciate it if the paper could provide more intuition for Lemma 1. Maybe a few equations would help.
>
> We thank you for this remark and will add, at least, the following example to the appendix (or directly to the paper if the page limit increases).
> Example: For $d = \partial_t - 1$ Lemma 1 results in solutions of the form $a\cdot\exp(t)$ for $a\in\mathbb{R}$. This one-dimensional solution space is described by the covariance function $k(t_1, t_2) = \exp(t_1+t_2)$.
>
> >This is minor, but I would like to see samples from the constrained Gaussian process, perhaps in one of Figures 2, 4, 5, or 6?
>
> The figures are already rather condensed. Following your suggestion, we decided to include the samples in the appendix.
>
> >Why does only Figure 2 show the credibility intervals? The remaining figures only discuss the mean. Is there a specific reason?
>
> Figure 2 was used as an example to show how the credibility intervals behave for the running example of the paper, and similar figures for the other systems exist in the Appendix.
> The other figures are supposed to show either the original data as in Figures 5 and 6, or highlight a specific behaviour of the LODE-GP, as in Figure 4.
> But thanks to your comment we have added a remark in each experiment that an example GP is shown in the respective appendices.
>
> **Regarding your questions:**
>
> >Abstract: what does "novel algorithmic and symbolic construction" mean? It is written frequently, but I don't understand "algorithmic and symbolic construction".
>
> Algorithmic: We can write down the steps to actually calculate all of the components and can also translate the proof for Theorem 1 directly to an algorithm, which is our code. Everything is done fully automated by the code.
> Symbolic: We use purely symbolic methods to create the covariance functions.
>
> >Line 87: Please explain the following notation in the paper R⪯. I suppose it refers to some sort of positive-definiteness?
>
> Thx, we made the notation clearer.
>
> >Line 223f: "We additionally analyse the training runtime and see that...": are those results in some figure/table? I can't seem to find the results
>
> The current (not the older one in the supplementary) version of our implementation of LODE-GPs has a speed comparable to a standard GPyTorch GP (8.82 seconds vs. 7.67 seconds on 50 datapoints for 300 training iterations). We currently prepare a proper time analysis with a runtime table in the appendix of the camera ready version.
>
> **Regarding your stated Limitations:**
> We would argue that the first time the constraints of the model are stated (i.e. in the Abstract in line 4), we precisely define the context of our model as: *This paper presents a novel algorithmic and symbolic construction for covariance functions of Gaussian Processes (GPs) with realizations strictly following a **system of linear homogeneous ODEs with constant coefficients**, which we call LODE-GPs.*
> This precise statement is repeated in the introduction (lines 47-49) and before Theorem 1 (line 140).
> We hoped that this would be early and clear enough for the readers and read all successive occurences as variations of the same statement with a focus on the currently most important restriction.
> Nevertheless, we are very open to suggestions on how we can make our constraints clearer throughout the paper.
>
> Finally, we thank you again for your review and the valuable feedback that resulted in modifications in the paper.

---

### Official Review · Reviewer_s3hy · 2022-07-13

**Rating:** 7
**Confidence:** 3
**Soundness:** 3 good
**Presentation:** 3 good
**Contribution:** 3 good

**Summary:**

The paper presents a way to algorithmically construct Gaussian Process
covariance functions in such a way, that realizations strictly follow a given
linear ordinary differential equation. The proposed construction via smith
normal form algorithms overcomes existing limitations on controllability of the
system, which exists in other approaches. The resulting LODE-GPs are compared to
classic GPs in three experiments.


**Questions:**

My most relevant question / suggestion is about the relation to parameter inference problems, stated above under "weaknesses".

Some questions / comments on probabilistic numerics (PN):
- l. 23 writes that probabilistic ODE solvers allow to "learn" non-linear ODEs,
  but it is not quite clear to me in which sense this holds (i.e. inferring the
  solution or inferring parameters). To my understanding, those methods "solve"
  or "estimate the solution of" ODE initial value problems (IVPs)
- Solving IVPs with non-linear ODEs _requires_ approximations, since they can, in
  general, not be solved in closed form. It might thus be misleading to state
  approximations as limitations of the solver, rather than a necessity of the
  general problem. In contrast, IVPs with linear ODEs can be solved in closed
  form, and therefore PN does not apply since no numerical error needs to be
  estimated.
- l. 70: What is meant with "complex calculations"? It could be helpful to
  clarify.
- Page 6, footnote: What is meant with "free functions in their solution set"?
  Depending on the meant object, [47] might possibly be in disagreement since it
  describes inference of time-varying system parameters (= unknown functions).

I want to highlight that these comments are only meant as helpful suggestions and not as necessary changes.
The resolution about the relation to parameter inference, commented on above, as
well as possibly the extension of the baseline for the experimental evaluation
are of higher relevance.

Minor comments:
- l. 18: "such _as_ in"
- l. 88 contains a lengthscale l, but it is not mentioned.
- Around Lemma 1 it would be helpful to point the reader to the proof in the
  appendix.


**Limitations:**

The authors adequately discussed the limitations of the proposed method.

**Strengths And Weaknesses:**

# Strengths
The presented method appears to be novel and I believe that it is a significant
contribution to research on Gaussian processes and generally machine learning in
the context of dynamical systems. The technical statements in the paper appear
sound. Related work is adequately cited (some comments below). Additionally, the
paper is very well-written and the method is presented in a clear way; the
bipendulum example used throughout the paper is very helpful, and the
experimental setup is discussed in much detail in the appendix.

# Weaknesses
The main criticism I have for the paper is about the comparison only to a
classic Gaussian process in the experiments. While I believe that this _might_ be
appropriate, it would be helpful to comment more extensively to why this is the
only reasonable comparison, and to define the problem setting more formally /
precisely.

For example, the experiment settings in section 5 can also be interpreted as
parameter / initial value inference problems, which could be approached with
e.g. least-squares regression with classic ODE solvers (e.g. [Biegler]) or
probabilistic numerics [Tronarp], or (probabilistic) gradient matching (e.g.
[Wenk]). Note that the former two approaches would further come with the
advantage of having linear O(n) scaling (due to not modeling covariances or to
using Gauss--Markov priors, respectively). Clarfying the problem setting further
could be helpful for readers to understand why these do not apply; otherwise
some of these approaches might be relevant baselines.

[Biegler] Nonlinear parameter estimation: A case study comparison, Biegler et al, 1986
[Wenk] ODIN: ODE-informed regression for parameter and state inference in time-continuous dynamical systems, Wenk et al, 2020
[Tronarp] Fenrir: Physics-Enhanced Regression for Initial Value Problems, Tronarp et al, 2022

---

> ### Author Response · Authors · 2022-08-02
> **Response to Reviewer s3hy**
>
> Thank you for your well written review to our paper and the helpful comments you made. We gladly incoporate them to improve our paper.
>
> **Regarding the comparison to only classic GPs**:
> The LODE-GPs can indeed be used to solve tasks formulated as an initial value problem (IVP).
> The works you have cited from Biegler, Wenk and Tronarp all focus on systems of nonlinear differential equations in their experiments, which can't be tackled with our approach.
> Vice versa, their approaches make the (typical for IVPs) assumption that the solution spaces are of finite dimension (take some suitable variant of "dimension", since the solution space is not a vector-space).
> This is often the case if you have as many differential equations as functions, but not in any of our three experiments.
> In our setup, the additional (infinite) degrees of freedom come from the $d_i = 0$ (corresponding to the Squared Exponential kernel).
>
> Of course, there are many systems both satisfying our approaches and those of Biegler, Wenk, or Tronarp.
> Important examples are systems of the form $f'=Af$ for a real-valued square matrix $A$ or a single differential equations of the form $f''+a\cdot f'+b\cdot f$.
> These systems have a finite dimensional solution (vector-)space and our approach of LODE-GPs simplifies to Bayesian linear regression in this solution space.
>
> Nevertheless, your comments caused an extended discussion of the task and its limitations in the revision.
>
> **Regarding the parameter learning**:
> The possibility of learning parameters was more of a nice side-effect so far, you are right that it certainly is an interesting topic to explore.
> We would argue that it is out of the scope as it was never the main focus of our work.
> Still, this comparison would again suffer from the same issues as discussed above as we don't have a system at hand which is actually solvable by our and the cited approaches.
>
> **Questions**
>
> >l. 23 writes that probabilistic ODE solvers allow to "learn" non-linear ODEs, but it is not quite clear to me in which sense this holds (i.e. inferring the solution or inferring parameters). To my understanding, those methods "solve" or "estimate the solution of" ODE initial value problems (IVPs)
>
> Thank you for spotting this. We have replaced our imprecise wording with your suggested "estimate the solution of".
>
> >Solving IVPs with non-linear ODEs _requires_ approximations, since they can, in general, not be solved in closed form. It might thus be misleading to state approximations as limitations of the solver, rather than a necessity of the general problem. In contrast, IVPs with linear ODEs can be solved in closed form, and therefore PN does not apply since no numerical error needs to be estimated.
>
> First, some relevant systems of non-linear ODEs can be solved in closed form, e.g. using separation of variables or substitution, and require no approximation. However, as you state, current numeric methods necessarily approximate, amongst them the cited works, where [8, 49, 47] base their approach on Kalman filters and [48] uses Runge-Kutta methods. (One could argue whether these methods really approximate, since they converge against the correct solution in infinite time and when using infinite numerical precision.)
> However, we view this approximation as a drawback since they also approximate the solutions of the subclass of systems of linear differential equations we considered in the paper.
>
> Nevertheless, we see that the wording in l.23 was misleading. Hence, we made the wording more precise to hopefully prevent such confusion in the future.
>
> >l. 70: What is meant with "complex calculations"? It could be helpful to clarify.
>
> This statement is supposed to be a comparison to the pushforward calculation using the SNF, which are comparably simple matrix row and column operations for the SNF and differentiation of simple kernel functions that have an exponential form.
> Whereas e.g. [49] even combine two likelihoods to perform their calculations.
> We have now added a "sometimes" to signify that not all the cited works have calculations as elaborate as [49].
>
>
> >Page 6, footnote: What is meant with "free functions in their solution set"? Depending on the meant object, [47] might possibly be in disagreement since it describes inference of time-varying system parameters (= unknown functions).
>
> It is a philosophical question whether to consider these time-varying system parameters are free functions. Our approach is only able to deal with those in the linear case (i.e. $a(t) + f(t)$) by treating them as additional functions instead of parameters. Since we do not consider changing parameters, but observable functions, this approach is not directly applicable.
>
> We also corrected the minor comments, thank you for them.
> Finally, we want to thank your for your extensive and valuable feedback.

---

### Meta-Review · Area_Chair_6Fqp · 2022-08-26

**Recommendation:** Accept
**Confidence:** Certain

**Metareview:**

The reviewers found this paper novel, significant for the community, and well written. All four reviewers recommended accepting the paper. I also appreciated the numerous illustrative examples in the paper. You addressed many of the remaining questions/concerns in your rebuttal and in the author-reviewer discussion. Please, go through the reviews once more for the camera-ready and take these into account.

**Award:**

No

---

### Decision · Program_Chairs · 2022-09-14

Accept